# OPTIMAL SCALING NEEDS OPTIMAL NORM

## ABSTRACT

Despite recent progress in optimal hyperparameter transfer under model and dataset scaling, no unifying explanatory principle has been established. For Adam and Scion optimizers, we discover that *joint* optimal scaling across model and dataset sizes is conditioned on a single invariant: the operator norm of the output layer. Across models with up to 1.3B parameters trained on up to 138B tokens, the optimal learning rate/batch size pair $(\eta^*, B^*)$ consistently has the same operator norm value — a phenomenon we term *norm transfer*. This constant norm condition is necessary but not sufficient: while for each dataset size, multiple $(\eta, B)$ reach the optimal norm, only a unique $(\eta^*, B^*)$ achieves the best loss. As a sufficient condition, we provide the first measurement of $(\eta^*, B^*)$ scaling with dataset size for Scion, and find that the scaling rules are consistent with those of Adam. Tuning per-layer-group learning rates also improves model performance, with the output layer being the most sensitive and hidden layers benefiting from lower learning rates. We provide practical insights on norm-guided optimal scaling and release our Distributed Scion (`Disco`) implementation with logs from over two thousand runs to support research on LLM training dynamics at scale.

## 1 INTRODUCTION

Recent advancements in the domain of Large Language Models (LLMs) have been largely driven by the principle of scale. Increasing model size and training dataset volume consistently yields more capable systems (Hoffmann et al., 2022; Kaplan et al., 2020), yet at an increasing computational cost. Consequently, achieving *optimal scaling* — a training regime where hyperparameters are optimally configured with growing scale — becomes a necessary step to push the model frontier further.

To address the challenge of hyperparameter tuning, several powerful yet disparate methods have emerged. Theoretically grounded frameworks like Maximum Update Parametrization ($\mu$P) (Yang et al., 2022) help transfer optimal hyperparameters with model scaling. Meanwhile, empirical scaling laws (Li et al., 2025) provide rules of thumb for setting hyperparameters optimally when theory is absent, such as with dataset size scaling. Yet, these approaches often feel like pieces of a puzzle, with a unifying principle for scaling across *both* model and dataset dimensions remaining elusive.

Recently, an emerging paradigm of norm-based optimization (Bernstein & Newhouse, 2024a; Pethick et al., 2025a) has offered a new lens through which to view training dynamics: it reframes optimization as a process that controls the operator norms of the model's weight matrices and gradient updates. This perspective enables monitoring of model properties during training, potentially revealing insights deeper than the loss curve alone. This raises a natural question: **can the norm-based perspective shed light on how to unify optimal model and dataset size scaling?**

In this work, we argue that the answer is yes. By tracking and analyzing layer norms across thousands of experiments, we have made several discoveries, summarized below:

- **Unifying invariant for optimal scaling.** The operator norm of the output layer $\|\boldsymbol{W}_{\text{out}}\|_{\text{RMS}\to\infty}$ (see Definition 2) for the optimal learning rate ($\eta$) and batch size ($B$) configuration has the same value — in other words, is invariant or "transfers" — with both model scaling (in width and depth) and dataset scaling (Fig. 2), as observed for both Scion and Adam optimizers (Appendix A.12). We refer to this phenomenon as *norm transfer*, and it provides a *necessary condition for optimality*. However, it is not sufficient, as multiple non-optimal $(\eta, B)$ pairs can reach the same optimal norm value (Fig. 3a).

- **Scaling rules for the Scion optimizer.** As a *sufficient condition for optimality*, we empirically measure the relationship between optimal learning rate $\eta^*$, batch size $B$, and dataset size $D$. The result is $\eta^*(B, D) \propto B^{0.62} \cdot D^{-0.56}$, matching the known square-root scaling rules for the Adam optimizer. We further find that the optimal batch size scales as $B^*(D) \propto D^{0.45 \pm 0.07}$, leading to $\eta^*(D) \propto D^{-0.28 \pm 0.07}$. For fixed $D$, one can trade off $\eta^* \leftrightarrow B^*$ via the $\eta \propto \sqrt{B}$ rule within a low-sensitivity region around the optimal norm (Fig. 3b). While the model performance is insensitive to this change, this freedom can be of computational advantage, allowing for training with larger batch sizes.

- **Optimal per-layer-group learning rate.** Performance can be improved by up to 6% in relative loss through additional per-layer-group tuning. We observe that a learning rate ratio $\eta_{\text{input}} : \eta_{\text{hidden}} : \eta_{\text{output}} = 1 : 1/8 : 1$ is consistently optimal across dataset sizes and batch sizes (Fig. 4). We also find the uniform $1 : 1 : 1$ layout to be close to the optimal one. Among layer groups, the output layer is the most sensitive to tuning, with sensitivity decreasing gradually for the hidden layers and then the input layer.

- **Distributed Scion/Muon and experimental logs.** To facilitate further research on large-scale training dynamics, we release `Disco`[1], a distributed implementation of the Scion/Muon optimizer compatible with modern parallelization strategies, along with norm logs from over two thousand training runs conducted for this study.

## 2 METHODOLOGY

### 2.1 BACKGROUND & TERMINOLOGY

Recently, a fundamental shift in the field of optimal scaling occurred with the work of Yang et al. (2024). It changed the focus from model parametrizations towards the norm perspective by showing that Maximum Update Parametrization ($\mu P$) (Yang et al., 2022) can be derived from a more fundamental principle: enforcing a *spectral condition* on the model weights and their updates during the training. We briefly explain the idea behind these concepts below.

$\mu P$ introduces theoretically grounded scaling rules for hyperparameters as a function of model width in order to ensure "maximal" feature learning in the infinite width limit. This way, the model is guaranteed to learn meaningful features while remaining stable as one scales up its size. As an important by-product, it was found that models with different widths, once parameterized within $\mu P$, all share the same optimal hyperparameters (e.g. learning rate) — therefore allowing for what is known as *zero-shot hyperparameter transfer*. This property has been extensively used for the past years to ensure optimal model scaling by tuning hyperparameters for a small (proxy) model, and then effortlessly transferring them to a larger one (Gunter et al., 2024; Dey et al., 2024; Meta AI, 2025; Zuo et al., 2025).

In turn, the spectral condition specifies bounds on the norms of weights and weight updates that are necessary to ensure feature learning. More formally:

**Definition 1** (Spectral condition). *Consider applying a gradient update $\Delta \boldsymbol{W}_\ell \in \mathbb{R}^{d_{\text{out}}^\ell \times d_{\text{in}}^\ell}$ to the $\ell$th weight matrix $\boldsymbol{W}_\ell \in \mathbb{R}^{d_{\text{out}}^\ell \times d_{\text{in}}^\ell}$ for a layer $\ell = 1, \ldots, L$. The spectral norms of these matrices should satisfy*

$$\|\boldsymbol{W}_\ell\|_* = \Theta\left(\sqrt{\frac{d_{\text{out}}^\ell}{d_{\text{in}}^\ell}}\right) \qquad \text{and} \qquad \|\Delta \boldsymbol{W}_\ell\|_* = \Theta\left(\sqrt{\frac{d_{\text{out}}^\ell}{d_{\text{in}}^\ell}}\right), \tag{1}$$

where $\|\boldsymbol{W}\|_*$ is the spectral norm, also equal to the largest singular value of $\boldsymbol{W}$, and $\|\boldsymbol{x}\|_{\text{RMS}} = \|\boldsymbol{x}\|_2/\sqrt{d}$. The symbol $\Theta$ is employed following the "Big-O" notation, indicating scaling behaviour (in this case, "constant"[2]) w.r.t. infinite width limit $d \to +\infty$. If conditions in Definition 1 are met, the zero-shot hyperparameter transfer is guaranteed and the model is being trained in the $\mu P$ regime.

Let us rewrite Definition 1 in a more "natural" way as:

---

[1] `https://anonymous.4open.science/r/disco_iclr2026-E11D`
[2] Formally, $f(x) = \Theta(g(x))$ if there are constants $A, B > 0$ such that $A \cdot g(x) \leq f(x) \leq B \cdot g(x)$.

$$\|\boldsymbol{W}_\ell\|_{\mathrm{RMS}\to\mathrm{RMS}} = \Theta(1) \qquad \text{and} \qquad \|\Delta\boldsymbol{W}_\ell\|_{\mathrm{RMS}\to\mathrm{RMS}} = \Theta(1), \tag{2}$$

where we follow Large et al. (2024) and introduce the core concept of this work:

**Definition 2** (Induced operator norm[3]). *Given a matrix $\boldsymbol{W} \in \mathbb{R}^{d_{\mathrm{out}} \times d_{\mathrm{in}}}$ and two normed vector spaces $(\mathbb{R}^{d_{\mathrm{in}}}, \|\cdot\|_\alpha)$ and $(\mathbb{R}^{d_{\mathrm{out}}}, \|\cdot\|_\beta)$, the "$\alpha$ to $\beta$" induced operator norm is given by:*

$$\|\boldsymbol{W}\|_{\alpha\to\beta} = \max_{\boldsymbol{x} \in \mathbb{R}^{d_{\mathrm{in}}}} \frac{\|\boldsymbol{W}\boldsymbol{x}\|_\beta}{\|\boldsymbol{x}\|_\alpha}. \tag{3}$$

The operator norms we are most interested in will be:

$$\|\boldsymbol{W}\|_{1\to\mathrm{RMS}} \coloneqq \max_j \|\mathrm{col}_j(\boldsymbol{W})\|_{\mathrm{RMS}}, \tag{4}$$

$$\|\boldsymbol{W}\|_{\mathrm{RMS}\to\mathrm{RMS}} \coloneqq \sqrt{d_{\mathrm{in}}/d_{\mathrm{out}}}\,\|\boldsymbol{W}\|_*, \tag{5}$$

$$\|\boldsymbol{W}\|_{\mathrm{RMS}\to\infty} \coloneqq \max_i d_{\mathrm{in}}\,\|\mathrm{row}_i(\boldsymbol{W})\|_{\mathrm{RMS}}, \tag{6}$$

where $\mathrm{row}_i(.)$ and $\mathrm{col}_j(.)$ denote the $i$-th row and $j$-th column of a matrix. In order to control the operator norms, Bernstein & Newhouse (2024a) derived *duality maps*, i.e. transformation rules of the gradients induced by a given norm. Applying these transformations not only keeps the gradient updates within the required bound (e.g. Eq. 2), but also ensures the steepest descent under the chosen norm (Bernstein & Newhouse, 2024b). For the norms in Eq. 4–6, the corresponding duality maps for the gradient $\boldsymbol{G}$ with singular value decomposition (SVD) $\boldsymbol{G} = \boldsymbol{U}\boldsymbol{\Sigma}\boldsymbol{V}^\top$ are:

$$\|.\|_{1\to\mathrm{RMS}}: \quad \mathrm{col}_j(\boldsymbol{G}) \mapsto \frac{\mathrm{col}_j(\boldsymbol{G})}{\|\mathrm{col}_j(\boldsymbol{G})\|_{\mathrm{RMS}}} \tag{7}$$

$$\|.\|_{\mathrm{RMS}\to\mathrm{RMS}}: \quad \boldsymbol{G} \mapsto \sqrt{\tfrac{d_{\mathrm{out}}}{d_{\mathrm{in}}}} \times \boldsymbol{U}\boldsymbol{V}^\top \tag{8}$$

$$\|.\|_{\mathrm{RMS}\to\infty}: \quad \mathrm{row}_i(\boldsymbol{G}) \mapsto \frac{1}{d_{\mathrm{in}}} \frac{\mathrm{row}_i(\boldsymbol{G})}{\|\mathrm{row}_i(\boldsymbol{G})\|_{\mathrm{RMS}}} \tag{9}$$

where the $\|.\|_{\mathrm{RMS}\to\infty}$ norm was added by Pethick et al. (2025a). Moreover, they wrapped the norm-based approach outlined above into a Scion optimizer.

Within Scion, one has to assign an operator norm to each layer, e.g. out of those in Eq. 4–6. The corresponding duality maps determine how raw gradients should be transformed for those layers before the optimizer updates the weights. For simplicity, layers are typically grouped as input, hidden, and output, and norms are assigned to these groups. Importantly, model weights are not explicitly transformed within Scion; only the weight updates are, via duality maps.

One prominent example of the norm-based view on model optimization is the Muon optimizer (Jordan et al., 2024), which proved to outperform Adam at scale (Liu et al., 2025; Wang et al., 2025) and showed great performance for models up to 1T parameters (Team et al., 2025). Muon can be viewed as a specific instantiation of Scion: it optimizes hidden layers under $\|.\|_{\mathrm{RMS}\to\mathrm{RMS}}$ assumption, and uses Adam for the remaining parameters. However, only in the case with no exponential moving average does Adam coincide with the steepest descent in "max-of-max norm" (Bernstein & Newhouse, 2024b). Since this is uncommon in practice, no "natural" norm applies, making Muon hard to analyze through the norm lens. By contrast, Scion naturally incorporates the norm perspective, updating every layer with an assigned, layer-specific norm, making it easy to interpret.

In practice, using norm-based optimizers as of now looks like a free lunch: they require only one momentum buffer[4] (compared to two for Adam), result in better performance with almost no computational overhead in large-scale distributed scenarios, and by design have zero-shot hyperparameter transfer built in. Moreover, the norm-based approach provides more insights into the dynamics of the model training: optimizer-assigned norms can be used naturally to monitor the training dynamics on a per-layer basis. This observation leads us to discoveries that we describe in Sec. 3.

---

[3]In the following we will omit "induced operator" for simplicity.

[4]Or even none, see `ScionLight` (Pethick et al., 2025a).

## 2.2 TRAINING SETUP

In all experiments, we use the Llama 3 architecture (Grattafiori et al., 2024) and `torchtitan` training framework (Liang et al., 2025). Most of the experiments are performed on the model with a total size of 69M trainable parameters (including input/output embedding layers), hereafter referred to as proxy model. For additional ablations in Sec. 3.2, we scale up the model up to $\times 12$ in width (to 1.3B parameters) and up to $\times 32$ in depth (to 168M). Notably, we employ a `norm-everywhere` approach, inspired by the concept of well-normedness in Large et al. (2024) and the recent line of work (Loshchilov et al., 2025; Kim et al., 2025). Effectively, we ensure that the input $x$ to every `Linear` layer is normalized to $\|x\|_{\text{RMS}} = 1$ by a preceding `RMSNorm` layer without learnable parameters. More details on model configurations are provided in Appendix A.2 and Appendix A.3.

As optimizer, we use Scion without weight decay (i.e. its unconstrained version) (Pethick et al., 2025a) without momentum and with the norm assumptions $\|.\|_{1 \to \text{RMS}} \Rightarrow \|.\|_{\text{RMS} \to \text{RMS}} \Rightarrow \|.\|_{\text{RMS} \to \infty}$ for input $\Rightarrow$ hidden $\Rightarrow$ output layers. Furthermore, we developed its distributed version, which natively integrates into `torchtitan`, supports FSDP/DDP/TP/EP/CP/PP strategies, and greatly speeds up the training at scale compared to the standard implementation. We make it openly available and provide more details in Appendix A.5.

For pretraining, we use a high-quality partition of the Nemotron-CC dataset (Su et al., 2025), Llama 3 tokenizer (Grattafiori et al., 2024) with a vocabulary size of 128,256 (after padding) and a context window of 4096. All the models are pretrained with the causal language modelling task. Unless stated otherwise, a constant learning rate schedule without warmup and without decay is used. This allows us, for a given set of hyperparameters, to perform a single long run and evaluate progressively larger dataset sizes, rather than conducting several runs for each dataset individually, thereby substantially reducing computational costs (Hu et al., 2024; Hägele et al., 2024).

## 2.3 OPTIMAL NORM MEASUREMENT

Our initial intuition was that for a given model and data scale, there is always some optimal norm value, corresponding to some optimal hyperparameter choice. To establish this, we focus on the output layer with the Scion-assigned $\|.\|_{\text{RMS} \to \infty}$ norm (hereafter referred to as *output norm*) as being the most natural layer to study.[5] The choice of $\|.\|_{\text{RMS} \to \infty}$ norm is motivated by Bernstein & Newhouse (2024a) as mapping from a "natural" continuous RMS norm semantics for hidden model representations onto a discrete vocabulary, although we also ablate this in Appendix A.15.2. Since by default we disable momentum and any regularization, we are only left with learning rate ($\eta$) and batch size ($B$) as hyperparameters to tune for optimality.

To extract the optimal hyperparameter configuration and the corresponding optimal norm, we run an ($\eta, B$) grid search for a given model and a given pretraining dataset size (hereafter referred to as horizon $D$, measured in tokens), and evaluate the model performance with training loss (cross-entropy of the next token prediction). Since we train in a non-repeating "infinite-data" regime, training loss faithfully reflects model performance and its generalization. First, we examine how the optimal norm, associated to a ($\eta^*, B^*$) configuration optimal for a given horizon, changes as the horizon increases. Then, we fix the horizon and scale up the model in width and depth, repeating the same optimal norm measurement. This way, we study both model and dataset scaling directions.

Practically, for every batch size we are interested in "marginalising" or "profiling" across learning rates, i.e. picking the optimal one and the corresponding output norm (see Appendix A.2 for details on the grid and random seed variations). However, an empirically lowest-loss point across the learning rate grid turned out to be a statistically noisy estimate; therefore, for each batch size, we perform a fit to the distribution of training loss vs. output norm across learning rates. Finally, we extract the optimal norm value from the fitted curve and the corresponding learning rate from the nearest data point to the fitted optimum. We provide more details on the fitting procedure in Appendix A.4.

---

[5]The output layer is invariant to both width and depth scaling, it is the most sensitive to learning rate tuning (Sec. 3.4), and it can be viewed as a linear classifier on the learned hidden representations. These considerations make us believe that the output layer plays a "representative" role for the entire model, thus making it a distinct layer to analyse.

## 3 RESULTS

### 3.1 OUTPUT NORM DYNAMICS

First, we describe how the output layer norm evolves depending on the hyperparameter settings. From learning rate scans, we observe that indeed there is an optimal norm value for a given batch size and horizon (Fig. 1a). Furthermore, learning rate is positively correlated with the output norm: the higher the learning rate, the higher the norm. Since we use an unconstrained version of Scion, the norms generally grow with the number of gradient steps (Fig. 1b and Appendix A.6). However, we note that norm values can also be constrained during training with weight decay (see Appendix A.9) or with various spectral clipping techniques (Newhouse et al., 2025). Intriguingly, the norm growth is not linear in log-log scale but *piecewise linear*: with the slope abruptly changing for all batch sizes at the norm value of $2^6 - 2^7$ and then at $2^9 - 2^{10}$, where for the latter the dynamics enters the "turbulence" region. This slope change may have the same nature as a recently observed phenomenon in the loss curve dynamics (Mircea et al., 2025). Last but not least, we observe that learning rate controls the " offset" of norm curves, and batch size controls the "decoupling degree" of curves: while early in training the curves of same $\eta$ but different $B$ are identical, the slope change at $2^6 - 2^7$ norm is more pronounced for larger batch sizes. Interestingly, after decoupling the curves seem to converge again to the same slope, that is lower than the initial one.

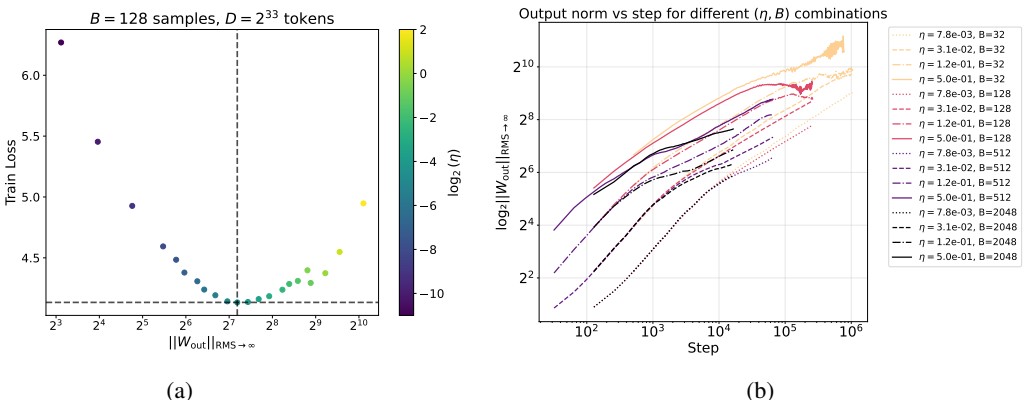

(a)                                                    (b)

Figure 1: **(a) Interplay of training loss, output layer norm $\|W_{\text{out}}\|_{\text{RMS}\to\infty}$ and learning rate**. Results are for the proxy model (69M parameters), batch size $B = 128$ samples and horizon $D = 2^{33}$ tokens. Points are colored by $\log_2(\eta)$ where $\eta$ is the learning rate. Black dashed lines mark the optimal configuration with minimum training loss. **(b) Growth of the output layer norm vs. gradient steps.** Each curve corresponds to a (learning rate $\eta$, batch size $B$) pair, with $B$ measured in samples; colour encodes batch size and line style encodes learning rate. See also the same plot vs. token horizons in Appendix A.6.

### 3.2 OPTIMAL NORM TRANSFER

After analysing learning rate scans across batch sizes, horizons and models of varying width/depth, we visualise results in Fig. 2, with an extended set of plots in Appendix A.7 and Appendix A.17. Each data point corresponds to optimally tuned learning rate $\eta^*$ for a given batch size, minimising training loss for that horizon and model. We report our observations below, separately for each direction of scaling, as well as additional ablations.

**Data scaling:** After profiling across learning rates and plotting optimal norm against batch size, we observe that for a given horizon there is a single optimal batch size with the corresponding optimal output norm $\|W_{\text{out}}\|_{\text{RMS}\to\infty} = 2^{7.0\pm0.2}$. Intriguingly, this norm value transfers across horizons. We refer to this phenomenon as *norm transfer*: the optimal $(\eta, B)$ configuration for a given horizon must result in the optimal norm of $\approx 2^7$. Also note that the optimal batch size grows with horizon scaling, which we discuss in Sec. 3.3. Interestingly, we observe the same norm transfer behavior when switching to a different dataset (Appendix A.11): specifically, Thai and Russian language partitions of Fineweb-2 (Penedo et al., 2025).

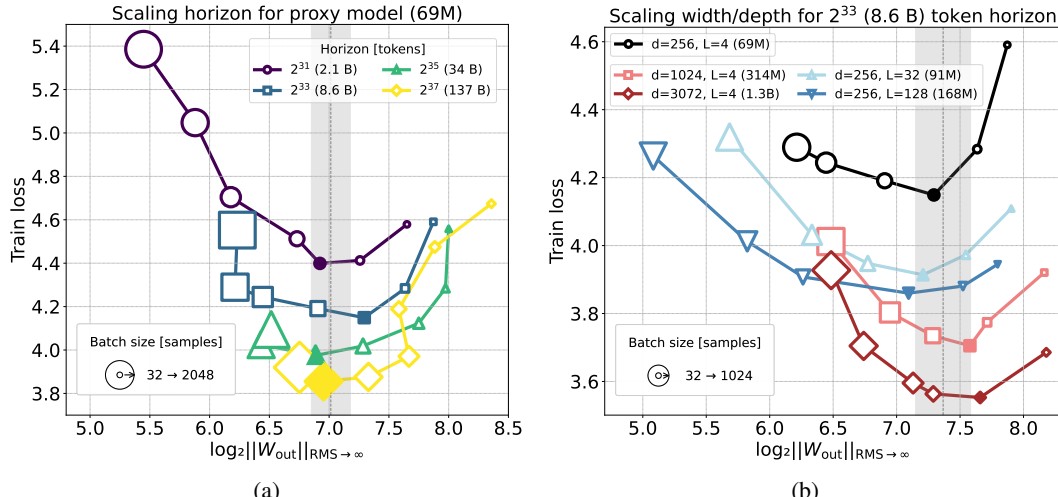

(a)                                                     (b)

Figure 2: **Training loss vs. output layer norm across batch sizes. (a)** Fixed proxy model (69M parameters) while increasing token horizon from $2^{31}$ to $2^{37}$. **(b)** Fixed token horizon $2^{33}$ while scaling width/depth of the proxy model as indicated in the legend. Each batch size point (increasing from 32 in $\times 2$ steps, reflected by marker size) has its learning rate optimally tuned. The optimal batch size per horizon/model configuration is indicated by the filled marker. All curves share optimal norm at $7.0 \pm 0.2$ across horizons and $7.4 \pm 0.2$ across models (grey band).

**Model width scaling:** It is expected to preserve the optimal norm by the design of our optimizer via the spectral condition (Eq. 1). Indeed, in Fig. 2b we observe that scaling up in width by a factor of $\times 12$ while keeping the horizon fixed results in the nested "$\mu P$-style" curves, sharing the same optimal norm while resulting in lower loss as we scale up.

**Model depth scaling:** Although not obvious *a priori*, we observe experimentally that scaling up in the number of layers by a factor of $\times 32$ results in norm transfer. This is quite surprising, since we do not employ any of the established depth-transfer techniques (Bordelon et al., 2023; Yang et al., 2023; Dey et al., 2025). We ablate them in Appendix A.10 and find that in our setup they all induce learning rate transfer, but our strategy (no residual scaling factors, initialization rescaling of layers prior to residuals by $1/\sqrt{2N_{\text{layers}}}$) results in the lowest loss. We speculate that this may be related to our `norm-everywhere` approach (Sec. 2.2) and uniformity in norm treatment by the optimizer and weight initialization.

**Momentum & learning rate decay:** In practice, one is more interested in Scion with non-zero momentum and with a decaying learning rate schedule as resulting in better performance. We study the impact of these two options in Appendix A.15 and observe that they both show norm transfer. Notably, the addition of momentum largely reduces sensitivity to batch size choice with multiple values resulting in the same optimal norm and loss (Fig. 14). The same is applicable to learning rate decay, which reduces sensitivity to learning rate choice (Fig. 18b).

**Adam optimizer:** As the optimizer commonly used in practice, we study its data scaling for the proxy model with two configurations: with momenta ($\beta_1 = 0.9, \beta_2 = 0.95$) and without ($\beta_1 = \beta_2 = 0$), where the latter case corresponds to a sign gradient descent. Intriguingly, for both we observe a clear norm transfer (Appendix A.12): the case without momentum exhibits the same optimal norm value ($\approx 2^7$) as the analogous without-momentum Scion, while the case with momentum has a noticeably higher optimal norm ($\approx 2^{11}$). We believe that this shared norm transfer pattern further supports a common norm-based view on optimization by Bernstein & Newhouse (2024b), and therefore makes our observations for Scion also transferable to Adam.

**Normalization layers:** Since we are interested in the norm structure of model scaling, our choice of `norm-everywhere` strategy may play a key role in the observed phenomena. We ablate various ways to place normalization layers (`RMSNorm` without trainable parameters) within the architecture in Appendix A.14. For the proxy model, fixed data horizon of 43B tokens, and fixed batch size $B = 256$ sequences we perform a learning rate scan while removing specific normalization layers

(see Appendix A.14 for a detailed description). In Fig. 13b for the case of the Scion optimizer with momentum $\alpha = 0.1$ we observe that removing normalization in residual connections results in significant training divergences. The setup with QK-norm + residuals + output layer normalization results in the same learning rate profile as our default `norm-everywhere`, indicating redundancy of MLP- and VO-normalization. Furthermore, residuals + output configuration results in slightly better performance, albeit with higher learning rate sensitivity. Likewise, addition of QK-norm largely reduces learning rate sensitivity, as also shown by Wortsman et al. (2023), thus making the training less sensitive to learning rate choice.

Finally, we selected the residuals-only configuration as the best trade-off between the minimalist usage of normalization layers, learning rate sensitivity, and model performance, and studied its norm scaling properties. Fig. 12 shows that norm transfer is present also in this scenario, interestingly with significantly lower optimal norm comparing to the `norm-everywhere` setting ($\approx 2^3$, one order of magnitude lower). One can also notice from Fig. 13b (see norm values in the legend) that it is the removal of the output layer norm that induces this reduction. This finding poses an interesting question: can it be in any way advantageous to prefer the model with lower weight norms? Intuitively, we would answer positively, referring to discussions in Newhouse et al. (2025), but leave detailed studies for future work.

> ↪ **Summary I:** Within the Scion framework, optimal norm transfers in both model (width and depth) and data scaling directions: it is *necessary* to choose the hyperparameter configuration so that the model output norm $\|\boldsymbol{W}_{\text{out}}\|_{\text{RMS}\to\infty}$ falls into the optimal region. Substituting alternative norms ($\|\boldsymbol{W}_{\text{out}}\|_{\text{RMS}\to\text{RMS}}$ or $\|\boldsymbol{W}_{\text{in}}\|_{1\to\text{RMS}}$) maintains the transfer consistency. The same behaviour holds with non-zero momentum, learning rate decay, and residuals-only layer normalization, as well as for the Adam optimizer. The optimal norm value decreases by a factor of 10 with the removal of the normalization layer before the model output layer.

### 3.3 Optimal $(\eta, B)$ scaling rule

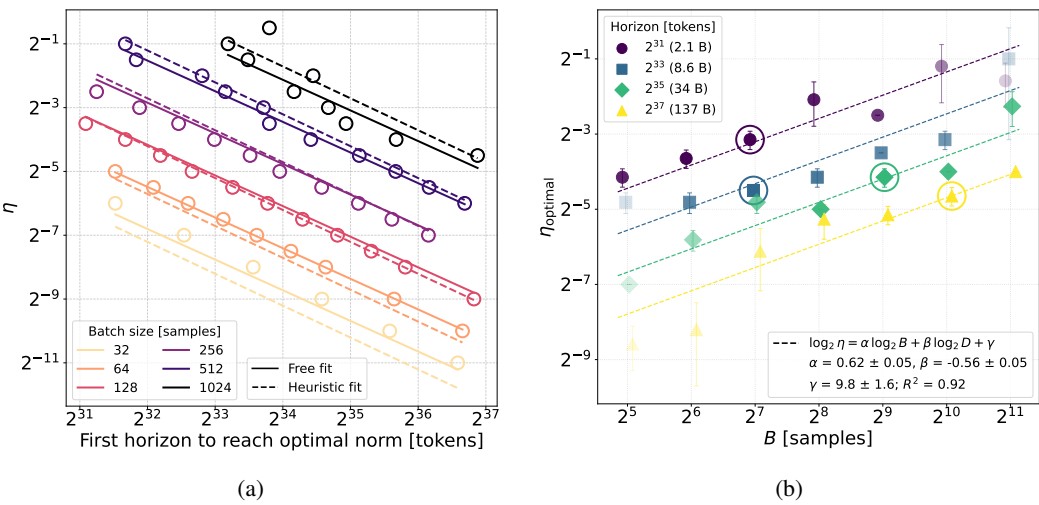

(a)  (b)

Figure 3: **(a) $(\eta, B)$ combinations that reach the optimal norm $\|\boldsymbol{W}_{\text{out}}\|_{\text{RMS}\to\infty} = 2^{7.0\pm0.2}$ for a given token horizon.** Colours denote batch size ($B$); the y-axis is learning rate ($\eta$). Solid and dashed lines denote free and heuristic fits (described in text). **(b) Optimal learning rate per batch size across horizons.** Circled markers indicate optimal $(\eta^*, B^*)$ with the lowest loss. Within a horizon, marker transparency linearly interpolates between the lowest- and highest-loss runs, with higher transparency indicating higher training loss. Error bars show systematic variation from the fitting method (Appendix A.4). Dashed lines are a joint linear regression with $\log_2 \eta^* \sim \log_2 B + \log_2 D$.

Despite the discovered norm guidance, it is still not obvious how to select the corresponding optimal combination of learning rate and batch size for a given horizon. Or more generally, what is the *sufficient* condition for optimality? In this Section, we explore this question.

Fig. 3a illustrates that the optimal norm condition observed in Fig. 2 is necessary but not sufficient. For each token horizon (x-axis), we plot the learning rates (y-axis) and batch sizes (colour) that reach[6] the optimal-norm region $\|\boldsymbol{W}_{\text{out}}\|_{\text{RMS}\to\infty} \in [2^{6.8}, 2^{7.2}]$. One can observe that for a given horizon, every batch size will reach optimal norm with a sufficiently high learning rate. We fit the data with linear models $\log_2 \eta = \alpha_{\text{first}} \log_2 B + \beta_{\text{first}} \log_2 D_{\text{first}} + \gamma_{\text{first}}$ (free fit) and $\log_2 \eta = 1.5 \log_2 B - \log_2 D_{\text{first}} + \gamma_{\text{first}}$ (heuristic fit). For the free fit, we find the exponents $\alpha_{\text{first}} = 1.32 \pm 0.03$ and $\beta_{\text{first}} = 0.96 \pm 0.03$, which are close to the values from the heuristic fit.

Hence, we cannot rely on the output norm as a guide to selecting optimal hyperparameters; it is only a necessary and not a sufficient condition. Let us now study sufficient conditions by first unfolding Fig. 2a and including optimal learning rate information that was profiled away. Specifically, we are interested in how the optimal learning rate $\eta^*$ changes within a fixed horizon $D$ with the batch size $B$ change, and then with horizons $D$ scaled up. Fig. 3b shows the corresponding data points along with a linear regression fit $\log_2 \eta^*(B, D) = \alpha \log_2 B + \beta \log_2 D + \gamma$. Note that only circled markers are per-horizon optima with the lowest loss. We observe several things:

- The coefficients of the fit $\alpha = 0.62 \pm 0.05$, $\beta = -0.56 \pm 0.05$ are consistent with a well-established square-root scaling with batch size (Malladi et al., 2024) and data horizon (Bjorck et al., 2025) for Adam, respectively. Similar to AI et al. (2025); Sato et al. (2025) we observe no surge phenomenon (Li et al., 2024), i.e. transition for a fixed $D$ from $\eta^* \propto \sqrt{B}$ to $\eta^* \propto 1/\sqrt{B}$ scaling rules for batch sizes higher than the critical one (Zhang et al., 2025). Theoretically, Jianlin (2025) explains this from the mean field theory perspective.

- Different batch sizes $B$ result in different losses, and for each horizon $D$ there is an optimal one $B^*(D)$, as emphasized in Fig. 3b with circled markers and marker transparency for relative loss difference. The optimal batch size increases with horizon scaling: in Appendix A.8 we measure with extended set of horizons $B^*(D) \propto D^{0.45\pm0.07}$, which is consistent with Adam (Li et al., 2025; Bergsma et al., 2025) and intriguingly with $B^* \propto \sqrt{D}$.

- Using $B^*(D) \propto D^{0.45}$ and $\log_2 \eta^*(B, D) \propto 0.62 \log_2 B - 0.56 \log_2 D$ with the corresponding uncertainties, we obtain for the optimal learning rate scaling $\eta^*(D) \propto D^{-0.28\pm0.07}$. This observation is consistent with Li et al. (2025) but appears to be in tension with Shen et al. (2024); Bergsma et al. (2025), albeit our methodologies are not fully comparable.[7] Again, this is interestingly close to $\eta^*(D) \propto D^{-1/4}$.

- Since there exists a single optimal batch size for each data horizon, the number of devices usable for training is fundamentally capped: beyond a point, increasing the number of devices either hurts throughput (small per-device microbatch size to keep the optimal global batch size) or degrades loss (leaving the optimal batch size region to keep throughput). This hints towards an interesting research direction: if this limit can be bypassed.

- In fact, for a fixed horizon, it is not a single optimal $(\eta^*, B^*)$ but an *optimal region* $(\eta^* \pm \Delta\eta, B^* \pm \Delta B)$ that results in near-optimality (opacity in Fig. 3b). We relate this to the notion of learning rate sensitivity (Wortsman et al., 2023) that we rephrase as *norm sensitivity*. We think this region is defined by the "flatness" of the horizon curve (Fig. 2a) around the optimal norm value. Within this region, one can "exchange" learning rate for batch size via the $\eta \propto \sqrt{B}$ rule, thus allowing for some flexibility in optimal hyperparameter choice, e.g. training with larger batch sizes.

---

↪ **Summary II:** For Scion, we measure the following hyperparameter scaling rules inducing the *sufficient* optimal scaling condition:

$$\eta^*(D) \propto D^{-0.28\pm0.07} \qquad \text{and} \qquad B^*(D) \propto D^{0.45\pm0.07}, \tag{10}$$

consistent with the Adam's scaling exponents. For a fixed horizon $D$, one can trade off $\eta^* \leftrightarrow B^*$ via the $\eta \propto \sqrt{B}$ rule within the region of low norm sensitivity, without loss in performance. By Scion's design, these observations hold true with model width scaling.

---

[6]Optimal norm will most likely be reached at some point (provided learning rate sweep resolution in Fig. 3a is too small), since in unconstrained Scion the weight norms are growing in time (see Sec. 3.1 and Fig. 1b).

[7]For example, because of weight decay usage in Bergsma et al. (2025), which significantly affects norm dynamics by constraining it, as we discuss in Sec. 5.

## 3.4 OPTIMAL PER-LAYER-GROUP LEARNING RATE

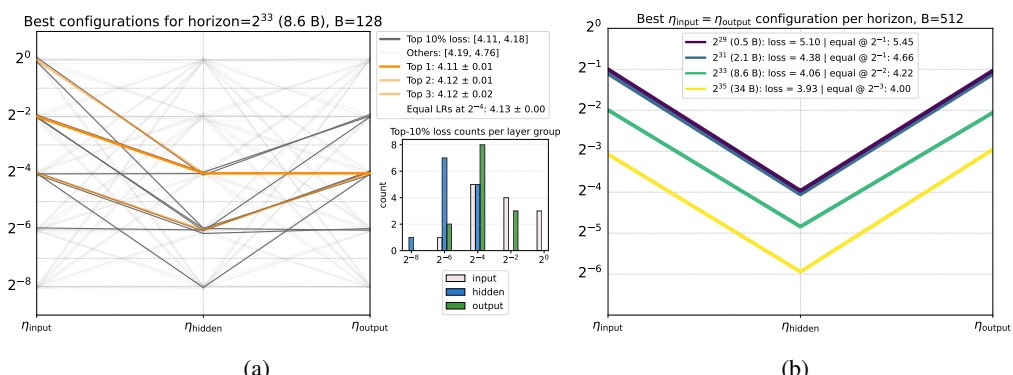

(a)                                    (b)

Figure 4: **(a) Parallel-coordinates view of per-layer-group learning rate tuning.** Results are for the proxy model (69M parameters) and batch size $B = 128$ samples, averaged across random seeds as described in Appendix A.2. Dark gray lines are the top 10% runs (loss 4.11–4.18); light gray lines are the remainder (loss 4.19–4.76). Orange traces highlight the three best settings. The inset histogram shows the distribution of top 10% counts for each layer group. **(b) Best learning rate layouts per training horizon under the constraint $\eta_{\text{input}} = \eta_{\text{output}}$.** Results are for the proxy model (69M parameters) and batch size $B = 512$ samples. All horizons favor a V-shaped layout with $\eta_{\text{hidden}}$ smaller than the input/output learning rates by the same $\times 1/8$ factor. In the legend we also report loss for the optimal $\eta_{\text{input}} = \eta_{\text{hidden}} = \eta_{\text{output}} \equiv \eta$ layout ("equal @$\eta$").

So far, we approached scaling from a "global" learning rate point of view. However, this may not be the case, and intricate dynamics can emerge where various layers require different learning rates at different scales to be trained optimally, thus questioning our conclusions so far. In this Section, we explore if this is the case.

Fig. 4a presents results for a proxy model (69M parameters), fixed data horizon (8.6B tokens) and fixed batch size ($B = 128$ samples, optimal for this horizon) where we run grid search over learning rate values $\eta \in \{2^{-8}, 2^{-7}, \ldots, 2^0\}$ for input (token embedding), output (linear projection onto vocabulary) and hidden (all the other) layers, averaged across random seeds (Appendix A.2). We observe that there is little optimal learning rate imbalance across layer groups, and uniform learning rate assignment results in the same loss as the optimal configurations within uncertainties. Furthermore, from the width of the optimal nodes count histograms per layer groups, we conclude that the output layer is the most sensitive to learning rate mistuning, with the sensitivity progressively decreasing for hidden and then input layers.

From analysing Fig. 4a and additional ones for different batch sizes (Appendix A.16) we found that the configuration $\eta_{\text{input}} : \eta_{\text{output}} : \eta_{\text{hidden}} = 1 : 1/8 : 1$ is always among the top 10%. This symmetry simplifies the learning scan and notably contradicts the optimal configurations suggested in Pethick et al. (2025a) and Riabinin et al. (2025). To study dynamics with horizon scaling, we perform the learning rate grid scan same as in Fig. 4a but with constraining $\eta_{\text{input}} = \eta_{\text{output}}$ [8], for the proxy model with $B = 512$. Fig. 4b illustrates the results, where we see the optimal hidden ratio ($\eta_{\text{input}}/\eta_{\text{hidden}} = 1/8$) transfer across horizons, as well as that it brings loss improvement w.r.t. a constant learning rate baseline. Lastly, we note that again, due to the optimizer design, we expect these observations to hold true under model width scaling.

---

↪ **Summary III:** Uniform learning rate configuration across layers is a strong baseline, which still can be improved with additional hidden layer group tuning: $\eta_{\text{input}} : \eta_{\text{output}} : \eta_{\text{hidden}} = 1 : 1/8 : 1$ yields a relative loss improvement of up to 6% and is transferable across dataset sizes.

---

[8]In terminology of Bernstein & Newhouse (2024a) this corresponds to *mass* tuning.

## 4 RELATED WORK

**Hyperparameters with model scaling** Yang et al. (2022) showed how to transfer optimal hyperparameters from a small to a large model in a principled way via Maximal Update Parametrization ($\mu$P). Everett et al. (2024) later showed that such transfer is also possible in other parametrizations. Yang et al. (2023); Dey et al. (2025) extended the method towards model scaling in depth. Empirically, scaling laws on how to set optimal hyperparameters as a function of compute (DeepSeek-AI et al., 2024), loss (Hu et al., 2024) or model size (Porian et al., 2025) were measured.

**Hyperparameters with data scaling** Remains poorly understood theoretically: Smith & Le (2018) showed for SGD how to adjust learning rate and batch size by modelling optimization trajectory as a stochastic differential equation (SDE). Largely, the problem has been approached experimentally by measuring hyperparameter scaling rules as a function of the dataset size (Shen et al., 2024; Hu et al., 2024; Filatov et al., 2025; Bergsma et al., 2025; Li et al., 2025).

$(\eta, B)$ **scaling rules** Historically, studies of interaction between learning rate and batch size emerged as an experimental effort to scale batch size without losing performance (Keskar et al., 2017; Goyal et al., 2018; Hilton et al., 2022). Later, a deeper understanding has been built from various theoretical angles: SDE (Malladi et al., 2024; Compagnoni et al., 2024), loss curvature (McCandlish et al., 2018), random matrix theory (Granziol et al., 2021).

**Norm-based optimization** Starting from the spectral condition (Yang et al., 2024), the approach of transforming gradient updates based on norm assumptions was fully established in Large et al. (2024); Bernstein & Newhouse (2024a), and recently explored in constraining weights themselves (Newhouse et al., 2025). The steepest descent view allowed for connections with manifold learning (Cesista, 2025) and optimizer design (Riabinin et al., 2025). This line of work has led to Muon (Jordan et al., 2024) and Scion (Pethick et al., 2025a;b), along with improvements (Ahn et al., 2025; Amsel et al., 2025), and benchmarks (Wen et al., 2025; Semenov et al., 2025) thereof.

## 5 CONCLUSION AND DISCUSSION

In this work, we demonstrate that the operator norm of the output layer is a powerful measure that guides joint optimal scaling across both model and dataset dimensions. Informally, we show:

1. $(\eta, B, D)$ choice $\xrightarrow{\text{affects}}$ layer operator norm (Sec. 3.1)
2. optimal loss $\xrightarrow{\text{requires}}$ optimal norm (Sec. 3.2)
3. optimal $\eta^*(D), B^*(D)$ scaling rules $\xrightarrow{\text{yield}}$ optimal loss (Sec. 3.3)

In words, we empirically (1) study how norms evolve with hyperparameter change and how to tune them to desired values; (2) demonstrate that the optimal hyperparameter configuration must have a predefined (output) layer norm in order to be transferable across data and model scales; (3) derive optimal hyperparameter scaling rules resulting in optimal loss.

While we are confident that the scaling rules in Sec. 3.3 hold at even larger scales, we still don't know why they are induced in this form, very much resembling square-root and 1/4-power laws. Moreover, how do these rules connect with our main finding, a necessary condition of scaling trajectory in (data, model) axes to have the same constant value — or one might say, to remain on a *manifold* (Bernstein, 2025). At this point more new questions arise:

- Why does optimal norm transfer? It is puzzling what makes the optimal scaling trajectory remain on the constant norm manifold, as well as what defines its structure.

- What is the reason behind optimal scaling rules? While we show how to set hyperparameters optimally, there is something missing in the norm perspective to explain it.

- Which norm is exactly optimal? We paid most of our attention to $\|\boldsymbol{W}_{\text{out}}\|_{\text{RMS}\to\infty}$, but are the observed phenomena really specific to this one only?

- How can the constant norm condition be leveraged? It looks like a naturally emerging inductive bias that one can take advantage of to optimize the training process.

We don't yet have answers to those questions, but we believe our study scratches the surface of exciting phenomena to be further understood.

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

## A Appendix

### A.1 LLM Usage

LLMs were used solely to aid in polishing the writing and improving the clarity of exposition. In addition, code-assistant tools were occasionally used for minor programming support, such as code completion and syntax suggestions; they were not employed to design algorithms, generate experiments, or implement the proposed methods from scratch.

### A.2 Model training configuration

- Proxy model, 69M parameters: 4 hidden layers with $d_{\text{model}} = 256$, Multi-Head Attention with $n_{\text{heads}} = 4$ and $n_{\text{kv-heads}} = 4$, SwiGLU activation function with MLP expansion factor $f_{\text{MLP}} = 2.75$, RoPE with $\theta = 10000$ (Su et al., 2024), Llama 3 tokenizer with vocabulary size of $128\,256$ (after padding) (Grattafiori et al., 2024), input and output embedding layers are not tied.

- $\times 4(12)$ wider model, 314M (1.3B) parameters: same as proxy, except $d_{\text{model}} = 1024$ (3072). In width scaling, we keep fixed $d_{\text{head}} = 64$ and scale the number of heads accordingly.

- $\times 8(32)$ deeper model, 91M (168M) parameters: same as proxy, except 32 (128) hidden layers.

- Semi-orthogonal initialization for hidden linear layers and row-wise normalized Gaussian initialization for input/output embedding layers (Pethick et al., 2025a). Initialisation of the last layer of both MLP and attention blocks (those with the output being added with the residual stream) is multiplied by $1/\sqrt{2N_{\text{layers}}}$.

- Dropout disabled, no biases in all `Linear` layers, no weight sharing between input and output embedding layers.

- `norm-everywhere`: normalise input to every `Linear` layer via `RMSNorm` without learnable parameters with $\epsilon = 1e^{-20}$. Effectively, this corresponds to Pre-LN setup with QK-norm plus three additional normalisation layers: V-norm, O-norm (before output projection matrix in Attention block), and MLP-norm (after SwiGLU and before the last MLP layer). Residual connections, including the ones injecting the input embedding layer information, remain intact.

- Random seeds:
  - For all proxy model runs in Sec. 3.2 and Sec. 3.3: `30`
  - For all width/depth-scaled-up model runs: interleaved `30 + 3034` (every $2^2$ step is `30`, every other $2^2$ step is `3034`)
  - For layout scans in Fig. 4a and Fig. 16: averaging over `30 + 3034 + 303409` for the three "core" learning rate values ($\{2^{-4}, 2^{-6}, 2^{-8}\}$ for $B = 32$, $\{2^{-2}, 2^{-4}, 2^{-6}\}$ for $B = 128$, $\{2^{-1}, 2^{-3}, 2^{-5}\}$ for $B = 512$), `3034 + 303409` for the rest
  - For layout scans in Fig. 4b: `30`

- `torchtitan` codebase, (Liang et al., 2025), FSDP2 (Feng et al., 2025), FlashAttention-2 (Dao, 2023)

### A.3 Optimizer configuration

Except dedicated ablations, we use the following set of hyperparameters:

- Unconstrained version (without weight decay),
- Learning rate $\eta$: grid with $2^{0.5}$ step for the proxy model, and $2^1$ step for the width/depth-scaled-up models,
- momentum $\mu = 0$, without Nesterov momentum,
- no warmup, constant learning rate schedule,
- $\epsilon = 1e^{-20}$ (used in gradient normalisation),

- orthogonalization of gradients for hidden layers ($\|.\|_{\text{RMS}\to\text{RMS}}$ norm assumption) with Newton-Schulz algorithm for $n_{\text{iter}} = 5$ with original Muon coefficients $a, b, c = (3.4445, -4.7750, 2.0315)$ (Jordan et al., 2024).

## A.4 OPTIMAL NORM FITTING & LOSS SMOOTHING

After naïvely taking the empirical optimum across the learning rate grid (e.g. as the one emphasized with dashed black lines in Fig. 1a), we found that the corresponding norm scans, although still indicating norm transfer, are quite noisy (e.g. compare Fig. 2a vs. Fig. 6a). From Fig. 1a we noted that data points in loss vs. norm plot resemble parabola if plotted in log-log scale. Furthermore, we know by design that at initialization (step 0) the output norm equals to 1, and train loss equals to 11.765. With this, we chose to perform a constrained fit with a second-order polynomial function in log-log scale $\log(\text{loss}) = a\log(\text{norm})^2 + b\log(\text{norm}) + c$, where the free term $c$ is fixed at precisely the loss value at initialization. We do this using weighted least squares fitting with `np.linalg.lstsq`, where the weighting is done with inverse uncertainties coming from *loss smoothing*, described below. For robustness, only seven data points around the empirical optimum are used in the fit. The optimal loss and norm values are then extracted as the parabola optimum coordinates. Optimal learning rate is taken from the data point closest to the fitted optimum. Results of such fits for Fig. 2 can be found in Fig. 17.

Since running several random seeds is computationally intensive, we perform *loss smoothing* to estimate the loss variance and make loss estimates more robust. Essentially, for a given horizon point, instead of taking its loss value, we average it with the previous and next evaluated points (67M tokens away, or e.g. 128 steps from each other with $B = 128$). Empirically estimated standard deviation is then used in the fits as described above. We employ loss smoothing only for small batch sizes $B \leq 128$, as those having large loss variance, and for large token horizons $D \geq 2^{33}$, to stay in the region of largely converged loss (which can be locally linearly approximated) and therefore not bias the estimate.

In order to get variance estimate in Fig. 3b without running several random seed runs, we vary the fitting procedure outlined above (with/without fitting, with/without loss smoothing, with/without constraint to loss at initialization), thus resulting in 6 total variations. For each of those we track how optimal norm/loss/learning rate changes, and propagate this variance to plotting and downstream analysis.

### A.5 DISTRIBUTED SCION

We implemented a distributed version of Scion/Muon. In this section, we briefly describe the implementation. We assume that the vectorized momentum buffer update is performed before applying the actual weight update.

#### A.5.1 DDP-DISCO

As a warm-up, we first consider the DDP case (note that a DDP-based version of Muon has already been implemented in `modded-nanogpt`[9]). Our implementation differs slightly from theirs, as we do not explicitly apply communication–computation overlap for DDP.

---

**Algorithm 1:** Disco `step_ddp`

---

**Input:** Parameters $\{p_i\}_{i=0}^{P-1}$ with $P = |\{p\}|$, world size $M$, local rank $r$

bucket_size $\leftarrow M$ ;

total_buckets $\leftarrow \lceil P/M \rceil$ ;

*global_updates* $\leftarrow$ array of length $P$ ;

```
/* Step 1:  Compute local updates                                  */
```
**for** $i = 0$ **to** $P - 1$ **do**
    **if** $i \bmod M = r$ **then**
        $g_i \leftarrow$ GETMOMENTUM$(p_i)$ ;
        $u_i \leftarrow$ LMO$(g_i)$ ;
        *global_updates*$[i] \leftarrow u_i$ ;

```
/* Step 2:  Communicate updates in buckets                         */
```
**for** $b = 0$ **to** *total_buckets* $-1$ **do**
    *start_idx* $\leftarrow b \cdot M$;
    *end_idx* $\leftarrow \min(start\_idx + M,\ P)$;
    *my_idx* $\leftarrow start\_idx + r$;
    **if** *my_idx* $<$ *end_idx* **then**
        $u_{\text{send}} \leftarrow$ *global_updates*$[my\_idx]$ ;
    **else**
        $u_{\text{send}} \leftarrow 0$
    $\{u_j\}_{j=0}^{M-1} \leftarrow$ ALLGATHER$(u_{\text{send}})$ ;
    **for** $j = 0$ **to** *end_idx* $-$ *start_idx* $-1$ **do**
        *global_updates*$[start\_idx + j] \leftarrow u_j$ ;

```
/* Step 3:  Apply updates vectorized                               */
```
APPLYUPDATES$(\{p_i\}_{i=0}^{P-1}, global\_updates)$ ;

---

**Helper functions:**

- GETMOMENTUM$(p)$: returns the momentum of $p$ from the momentum buffer.

- LMO$(g)$: runs the LMO based on the chosen norm of $p$.

- ALLGATHER$(u)$: gathers one tensor $u$ from each rank in the data-parallel group.

- APPLYUPDATES$(\{p\}, \{u\})$: applies the global updates $\{u\}$ to the parameters $\{p\}$ in a single vectorized operation.

Notice this version works out-of-the-box for PP+DDP, as we could let each PP(Pipeline Parallelism) stage only manage the parts of the model that the current PP stages needed for forward and backward.

To make it work with TP, one needs to do an extra all-gather in the local update loop.

---

[9]https://github.com/KellerJordan/modded-nanogpt

### A.5.2 FSDP-DISCO

Here, "FSDP" refers to a combination of `FSDP2` with arbitrary parallelisms, including Data Parallelism (DP), Context Parallelism (CP), Expert Parallelism (EP), Tensor Parallelism (TP), and Pipeline Parallelism (PP). In this section, we restrict our discussion to FSDP and EP (via DP2EP). In principle, there is no need to treat DP and PP separately: one only needs to all-gather the full gradient before communication in the FSDP case to ensure compatibility with TP.

We assume the design of this work, which applies an $\|.\|_{1\to\mathrm{RMS}}$ norm for the LLM's embedding layer and an $\|.\|_{\mathrm{RMS}\to\infty}$ norm for the output linear layer. (`SignNorm` is also acceptable and remains compatible if one strictly follows Scion's design.)

The `FSDP2` implementation in PyTorch shards weights and gradients along the tensor's first dimension. We discuss Disco under this assumption and further assume that each tensor or matrix corresponds to a single layer. Consequently, fused tensors such as `fused_QKV` in attention layers or `fused_W13` in SwiGLU are not supported.

Under these hypotheses, we can classify parameters into three groups: `embedding`, `experts`, and (pure-)`fsdp`. For updates, no extra communication is required for `embedding` and `experts` parameters, thanks to the `Shard(0)` strategy in `FSDP2`.

---

**Algorithm 2:** Disco `step_embedding`

---

**Input:** Embedding parameters $\{p_i\}_{i=0}^{P-1}$

```
/* Initialise updates storage                                  */
```
$updates \leftarrow$ array of length $P$ ;

```
/* get momentum and compute LMO update on local shards         */
```
**for** $i = 0$ **to** $P - 1$ **do**
  $g_i \leftarrow$ GETMOMENTUM($p_i$) ;
  $u_i \leftarrow$ LMO($g_i$) ;
  $updates[i] \leftarrow u_i$ ;

```
/* Apply updates vectorized                                    */
```
APPLYUPDATES($\{p_i\}_{i=0}^{P-1}$, $updates$) ;

---

---

**Algorithm 3:** Disco `step_experts`

---

**Input:** Expert parameters $\{p_i\}_{i=0}^{P-1}$, transpose flag $transpose$

```
/* Initialise updates storage                                  */
```
$updates \leftarrow$ array of length $P$ ;

```
/* get momentum and compute LMO update on local shards         */
```
**for** $i = 0$ **to** $P - 1$ **do**
  $g_i \leftarrow$ GETMOMENTUM($p_i$) ;
  $u_i \leftarrow$ BATCHEDLMO($g_i$; $transpose\_experts = transpose$) ;
  $updates[i] \leftarrow u_i$ ;

```
/* Apply updates vectorized                                    */
```
APPLYUPDATES($\{p_i\}_{i=0}^{P-1}$, $updates$) ;

---

Noting that MoE expert weights are typically laid out as either (total_experts, $d_{\mathrm{out}}$, $d_{\mathrm{in}}$) or (total_experts, $d_{\mathrm{in}}$, $d_{\mathrm{out}}$), we apply a transpose in the latter case to ensure that the output dimension comes first. In an FSDP + DP2EP setting, each gradient passed to `LMO` is therefore a 3D tensor with layout (local_experts, $d_{\mathrm{out}}$, $d_{\mathrm{in}}$). Accordingly, SVD or Newton-Schulz-based algorithms must correctly handle batched inputs.

And below is the algorithm for purely fsdp-shard parameters.

---

**Algorithm 4:** Disco `step_fsdp`

---

**Input:** FSDP-sharded parameters $\{p_i\}_{i=0}^{P-1}$, world size $M$ over `fsdp`, local rank $r$

bucket_size $\leftarrow M$;
total_buckets $\leftarrow \lceil P/M \rceil$;
*global_updates* $\leftarrow$ array of length $P$;

**for** $b = 0$ **to** *total_buckets* $-1$ **do**
    *start* $\leftarrow b \cdot M$;   *end* $\leftarrow \min(start + M,\ P)$;
    *my_idx* $\leftarrow start + r$;
    **for** $j = 0$ **to** $M - 1$ **do**
        $i \leftarrow start + j$;
        **if** $i < end$ **then**
            $g_i \leftarrow$ GETMOMENTUM$(p_i)$       // row-sharded by FSDP;
            *send_list*[j] $\leftarrow g_i$;
        **else**
            *send_list*[j] $\leftarrow 0$                  // zero padding

    *recv_list* $\leftarrow$ ALLTOALL(*send_list*)
    $g^\star \leftarrow$ CONCATROWS(*recv_list*)     // reconstruct full gradient for $p_{i^\star}$
    $u^\star \leftarrow$ LMO$(g^\star)$
    *updates_send_list* $\leftarrow$ SPLITROWS$(u^\star,\ M)$     // split $u^\star$ by rows;
    *updates_recv_list* $\leftarrow$ ALLTOALL(*updates_send_list*);
    **for** $j = 0$ **to** $end - start -1$ **do**
        *global_updates*[*start* + j] $\leftarrow$ *updates_recv_list*[j];

/* Single vectorized apply                                           */
APPLYUPDATES$(\{p_i\}_{i=0}^{P-1},$ *global_updates*$)$;

---

**Helper functions:**

- ALLTOALL(*list*): list-based ALLTOALL over `dp_shard_cp`.
- CONCATROWS(*list*): concatenates row-shards into a full tensor.
- SPLITROWS$(u, M)$: splits $u$ into $M$ contiguous row blocks.

## A.6 OUTPUT NORM EVOLUTION WITH DIFFERENT $(\eta, B)$

**Output norm vs horizon for different $(\eta, B)$ combinations**

(a)

**Output norm vs step for different $(\eta, B)$ combinations**

(b)

Figure 5: **Growth of the output layer norm $\|W_{\text{out}}\|_{\text{RMS}\to\infty}$ vs. horizon, in tokens (a) and number of steps (b).** Results are for the proxy model (69M parameters). Each curve is a (learning rate $\eta$, batch size $B$) pair, with $B$ measured in samples: colour encodes batch size and line style encodes learning rate, as described in the legend.

A.7   Supplementary plots to Fig. 2

Figure 6: **(a)** Fig. 2a with an extended set of horizons, raw data (i.e. no loss smoothing, no fitting, see Appendix A.4 for details on fitting and loss smoothing). **(b)** Same as (a) + loss smoothing. **(c)** Same as (a) + fitting. **(d)** Same as (a) + fitting + loss smoothing. **(e)** Fig. 2b, raw data (no loss smoothing, no fitting). **(f)** Same as (e) + loss smoothing.

## A.8 Optimal $B^*(D)$ measurement

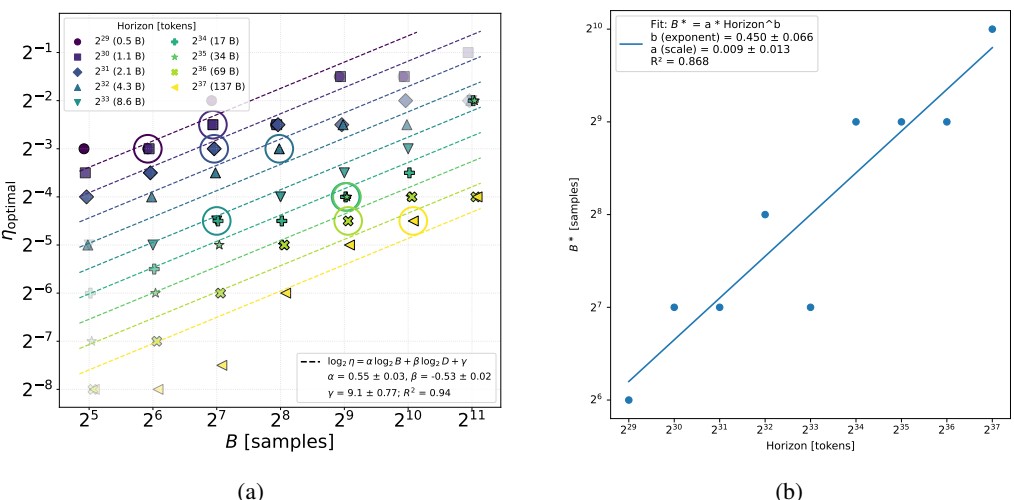

(a)                                                                                          (b)

Figure 7: **(a)** Same as Fig. 3b, but with extended set of horizons. **(b)** Optimal batch size $B^*$ vs. horizon, as extracted from (a)). The line is a power-law fit (described in legend).

Fig. 3b, for the sake of clarity and simplicity, illustrates only four horizons. This is not really sufficient to extract precisely the scaling of optimal $(\eta^*, B^*)$ (circled markers) with $D$, as it would mean fitting of four data points. We therefore perform the ordinary least squares (OLS) fit on the extended set of 9 horizons from Fig. 7a, effectively fitting the x-coordinate of the circled markers with a line. We model optimal batch size dependency on horizon $D$ as a power law $B^*(D) = aD^b$ and present results on Fig. 7b. We extract $B^* \propto D^{0.45\pm0.07}$, consistent with the square-root scaling.

## A.9 Norm constraint with weight decay

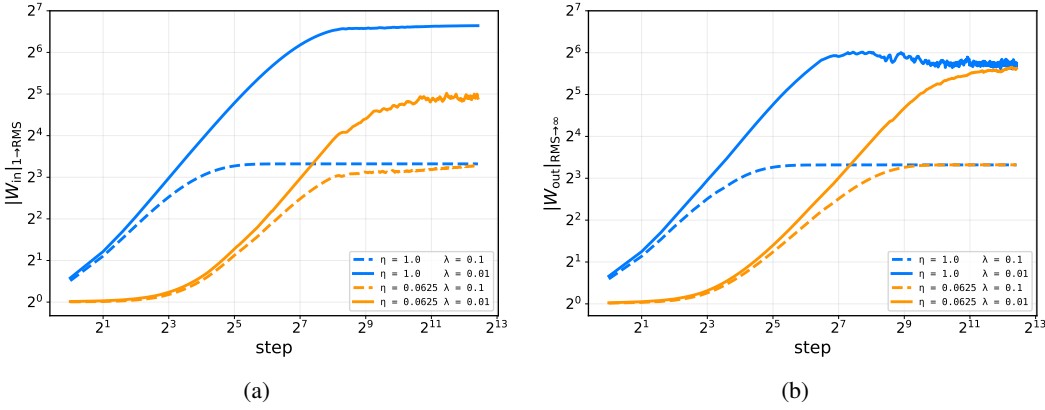

(a)                                                                                          (b)

Figure 8: **Operator norm against number of gradient update steps.** Fixed batch size $B = 32$, momentum $\mu = 0.1$, two values of learning rate $\eta = \{0.0625, 1.\}$ and two values of weight decay $\lambda = \{0.01, 0.1\}$ (applied as in Pethick et al. (2025a)), for a proxy model (69M parameters). **(a)** $\|W_{\text{in}}\|_{1\to\text{RMS}}$ norm **(b)** $\|W_{\text{out}}\|_{\text{RMS}\to\infty}$. We see for $\lambda = 0.1$ both norms converging to $1/\lambda$, while for $\lambda = 0.01$ asymptotic values are not conclusive.

## A.10 ABLATION OF DEPTH TRANSFER TECHNIQUES

Model: same as our proxy model (Appendix A.2), with the only difference in the head configuration: $n_{\text{query\_heads}} = 2$, $n_{\text{kv\_heads}} = 1$. We run a combination of two ablations: (i) weight initialisation depth-wise scaling (via gains/variance), and (ii) residual branch summation ratios.

For weight *initialisation*, the depth-wise scaling factors are applied to **only** the output linear projection of attention and SwiGLU. We compare three flavours of depth init scaling: `identity` (baseline), `total-depth`, and `relative-depth`, defined by multiplying the gain $\sigma$ by

$$\sigma* = \begin{cases} 1/\sqrt{2N_{\text{layers}}} & \text{scale by } \texttt{total-depth}, \\ 1/\sqrt{2\,l_i} & \text{scale by } \texttt{relative-depth}, \\ 1 & \text{scale by } \texttt{identity}. \end{cases} \quad (11)$$

where $N_{\text{layers}}$ is the total number of Transformer blocks, and $l_i \in \{1, \dots, 2N_{\text{layers}}\}$ is the relative depth of the current block; $\sigma$ is the scaled orthogonal gain, $\sigma = \sqrt{\frac{d_{\text{out}}}{d_{\text{in}}}}$, for hidden weights $W \in \mathbb{R}^{d_{\text{out}} \times d_{\text{in}}}$.

Each transformer block is assigned depth 2, since attention and FFN sub-blocks each count as depth 1. When using `relative-depth`, the depth of all FFN blocks can be offset by 1.

For depth-wise residual scaling, we write the residual connection in transformer as:

$$Y = \alpha \cdot X + \beta \cdot \text{Block}\big(\text{Norm}(X)\big), \quad (12)$$

where $X$ is the block input and $\text{Block}(\cdot)$ denotes either self-attention or a FFN, and $\text{Norm}$ is RM-SNorm in our setup.

We consider three depth-wise residual scaling schemes:

$$(\alpha, \beta) = \begin{cases} \left(\frac{2N_{\text{layers}}-1}{2N_{\text{layers}}}, \frac{1}{2N_{\text{layers}}}\right) & \text{scale by } \texttt{depth-normalized}, \\ \left(1, \frac{1}{2N_{\text{layers}}}\right) & \text{scale by } \texttt{completeP}, \\ (1, 1) & \text{scale by } \texttt{identity}. \end{cases} \quad (13)$$

`depth-normalized` Large et al. (2024) scales both the residual and block contributions proportionally to depth. `completeP` Dey et al. (2025) preserves the residual branch while scaling down the block contribution by depth. `identity` corresponds to the conventional unscaled residual formulation.

We fixed batch size ($B$) to 32 samples, the sequence length to 4096, and the number of training steps to 2048. Experiments were conducted using proxy models with depths $N_{\text{layers}} \in \{2, 16, 64\}$. For all models, we performed a sweep over the learning rate $\{2^{-4}, 2^{-3}, 2^{-2}, 2^{-1}, 2^0\}$.

We report the final-step losses in Table 1, Table 2, and Table 3 for the three depths, respectively, with the two lowest losses highlighted. From the perspective of learning rate transfer, we find that with our optimizer, the optimal learning rate consistently remains around $2^{-2}$, regardless of weight initialisation or residual scaling. We also observe that combining `total-depth` weight initialisation with `identity` residual scaling yields a negligible improvement compared to using `identity` weight initialisation.

Table 1: 2 layers ($B = 32$, steps=2048)

| Residual init | Residual multiplier | Learning rate $\eta$ | | | | |
|---|---|---|---|---|---|---|
| | | $2^{-4}$ | $2^{-3}$ | $2^{-2}$ | $2^{-1}$ | $2^{0}$ |
| total-depth | identity | 4.20 | **4.11** | **4.09** | 4.13 | 4.22 |
| total-depth | depth-normalized | 4.20 | **4.12** | **4.11** | 4.17 | 4.21 |
| total-depth | completeP | 4.22 | **4.15** | **4.16** | 4.17 | 4.28 |
| identity | identity | 4.19 | **4.10** | **4.10** | 4.13 | 4.23 |
| identity | depth-normalized | 4.22 | **4.12** | **4.12** | 4.13 | 4.21 |
| identity | completeP | 4.21 | **4.15** | **4.13** | 4.16 | 4.24 |
| relative-depth | identity | 4.20 | **4.11** | **4.09** | 4.13 | 4.23 |
| relative-depth | depth-normalized | 4.20 | **4.13** | **4.11** | 4.16 | 4.25 |
| relative-depth | completeP | 4.21 | **4.16** | **4.14** | 4.18 | 4.24 |

Table 2: 16 layers ($B = 32$, steps=2048)

| Residual init | Residual multiplier | Learning rate $\eta$ | | | | |
|---|---|---|---|---|---|---|
| | | $2^{-4}$ | $2^{-3}$ | $2^{-2}$ | $2^{-1}$ | $2^{0}$ |
| total-depth | identity | 3.81 | **3.75** | **3.73** | 3.77 | 3.88 |
| total-depth | depth-normalized | 3.85 | **3.79** | **3.80** | 3.84 | 3.92 |
| total-depth | completeP | 3.87 | **3.82** | **3.82** | 3.85 | 3.94 |
| identity | identity | 3.81 | **3.74** | **3.75** | 3.79 | 3.89 |
| identity | depth-normalized | 3.83 | **3.78** | **3.78** | 3.83 | 3.92 |
| identity | completeP | 3.86 | **3.81** | **3.81** | 3.85 | 3.94 |
| relative-depth | identity | 3.82 | **3.79** | **3.74** | 3.80 | 3.90 |
| relative-depth | depth-normalized | 3.84 | **3.79** | **3.80** | 3.83 | 3.95 |
| relative-depth | completeP | 3.88 | **3.82** | **3.82** | 3.85 | 3.95 |

Table 3: 64 layers ($B$=32, steps=2048)

| Residual init | Residual multiplier | Learning rate $\eta$ | | | | |
|---|---|---|---|---|---|---|
| | | $2^{-4}$ | $2^{-3}$ | $2^{-2}$ | $2^{-1}$ | $2^{0}$ |
| total-depth | identity | 3.67 | **3.60** | **3.60** | 3.65 | 3.79 |
| total-depth | depth-normalized | 3.71 | **3.65** | **3.65** | 3.69 | 3.80 |
| total-depth | completeP | 3.72 | **3.67** | **3.67** | 3.72 | 3.82 |
| identity | identity | 3.70 | **3.63** | **3.62** | 3.66 | 3.78 |
| identity | depth-normalized | 3.70 | **3.64** | **3.64** | 3.69 | 3.80 |
| identity | completeP | 3.70 | **3.70** | **3.67** | 3.72 | 3.82 |
| relative-depth | identity | 3.70 | **3.61** | **3.61** | 3.67 | 3.82 |
| relative-depth | depth-normalized | 3.71 | **3.65** | **3.65** | 3.69 | 3.80 |
| relative-depth | completeP | 3.72 | **3.68** | **3.67** | 3.73 | 3.83 |

### A.11 ABLATION WITH FINEWEB-2 DATASET

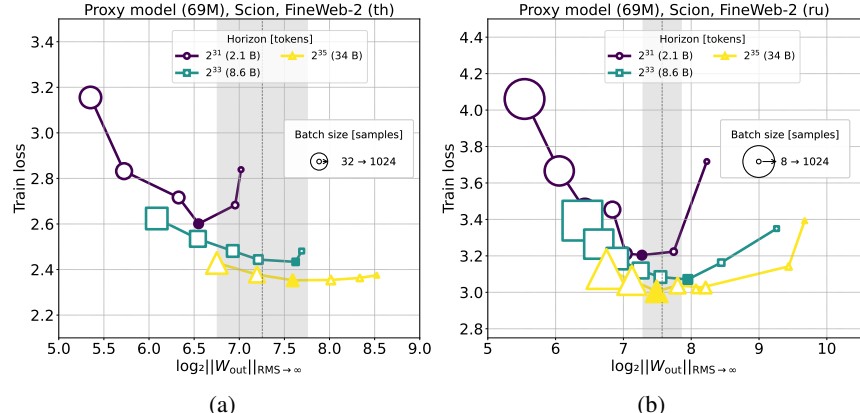

(a)          (b)

Figure 9: **Same as Fig. 2a but using the FineWeb-2 dataset:** **(a)** Thai partition, **(b)** Russian partition. We note that while for the Russian language the three horizon curves are nested inside of each other and share the same optimal norm, for the Thai partition the first horizon is off. This may be due to being an early phase of training or statistical fluctuation. For completeness, we provide the individual learning rate scans and fits used to produce this plot in Fig. 10.

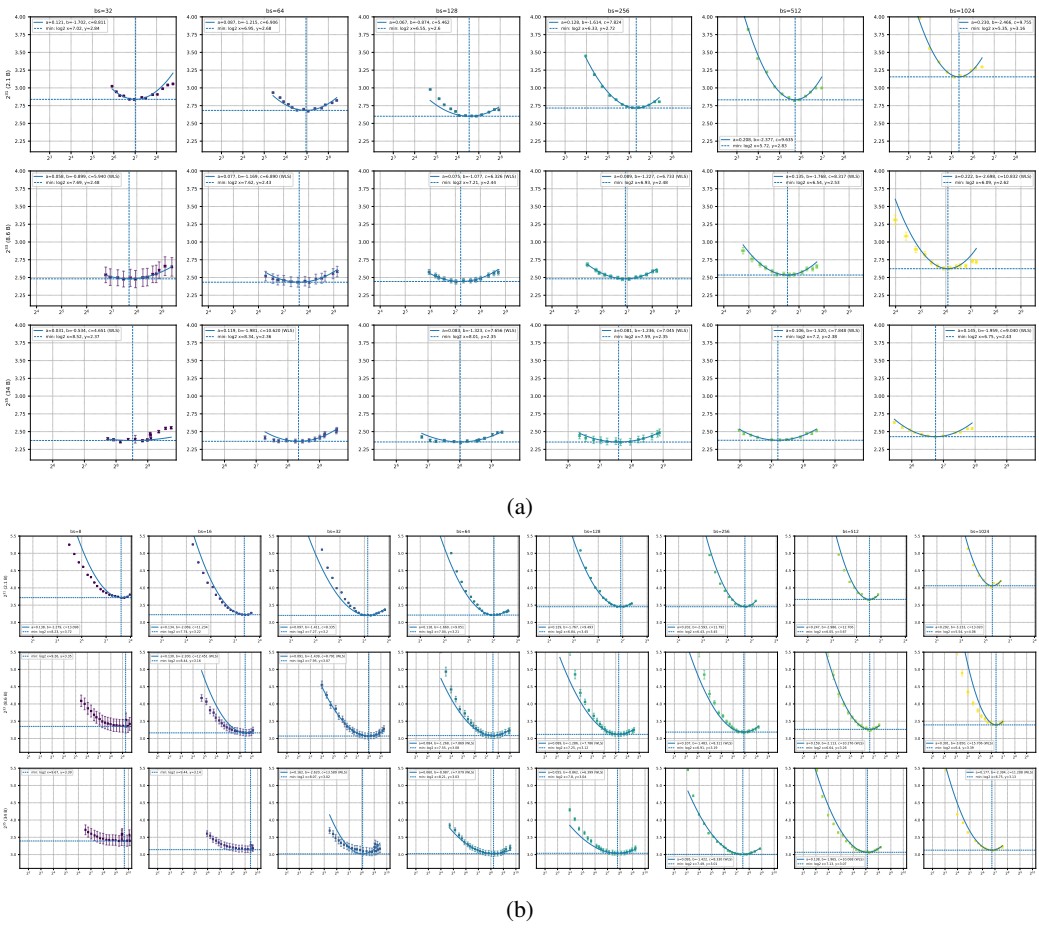

(a)

(b)

Figure 10: **Individual** $\|W_{\text{out}}\|_{\text{RMS}\to\infty}$ **norm scans for various batch sizes** $B$ **(columns) across various horizons** $D$ **(rows), for the FineWeb-2 dataset:** **(a)** Thai partition, **(b)** Russian partition.

## A.12 Ablation with Adam optimizer

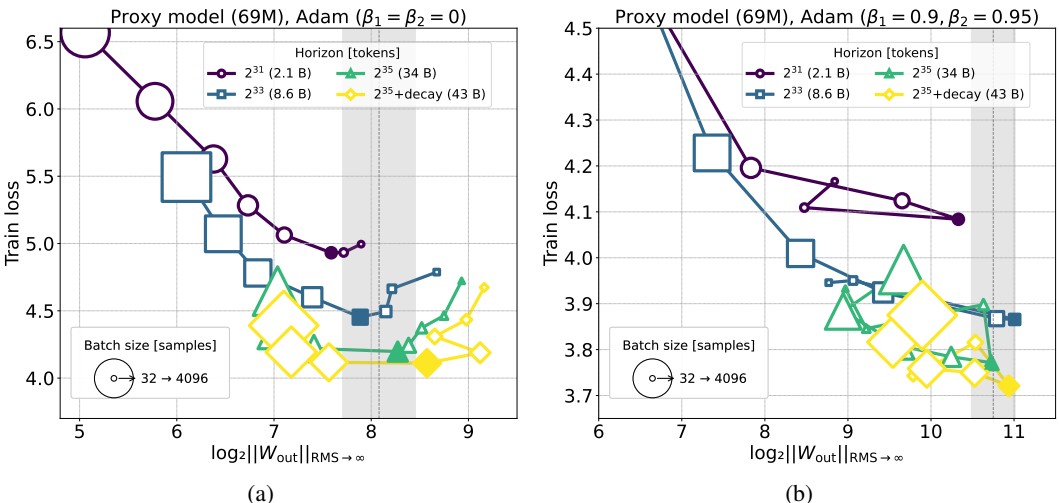

(a)                                                                     (b)

Figure 11: **Same as Fig. 2a but using the Adam optimizer** (no weight decay, warmup for $2^{29} \approx$ 537M tokens across all batch sizes, followed by constant learning rate schedule with linear decay to 0 for 20% of the total schedule): **(a)** $\beta_1 = \beta_2 = 0$, **(b)** $\beta_1 = 0.9$, $\beta_2 = 0.95$. For the no-momentum version (a) we observe norm transfer at the same optimal norm value as our main experiment with Scion (Fig. 2a). For the version with momentum (b), norm transfer is also present (with reduced sensitivity to batch size choice similarly to Scion), but notably at a higher optimal norm value.

## A.13 Residuals-only data scaling

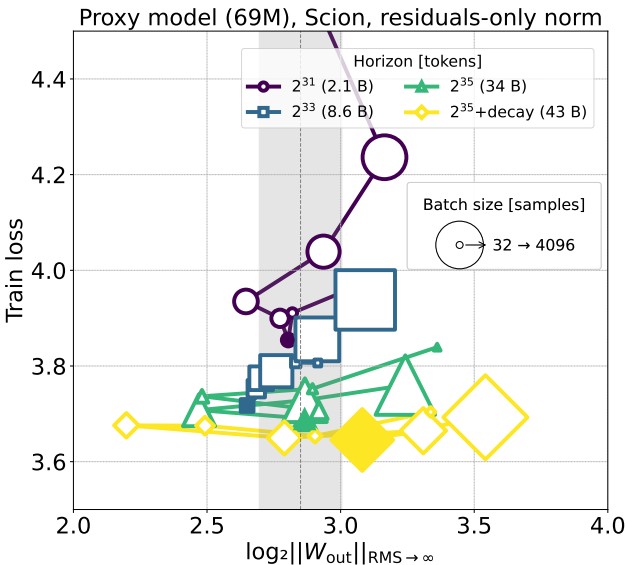

Figure 12: **Same as Fig. 2a but leaving in the proxy model only residuals normalization layers** (`RMSNorm` without trainable parameters normalizing inputs to every attention and MLP blocks).

## A.14  NORMALIZATION LAYERS

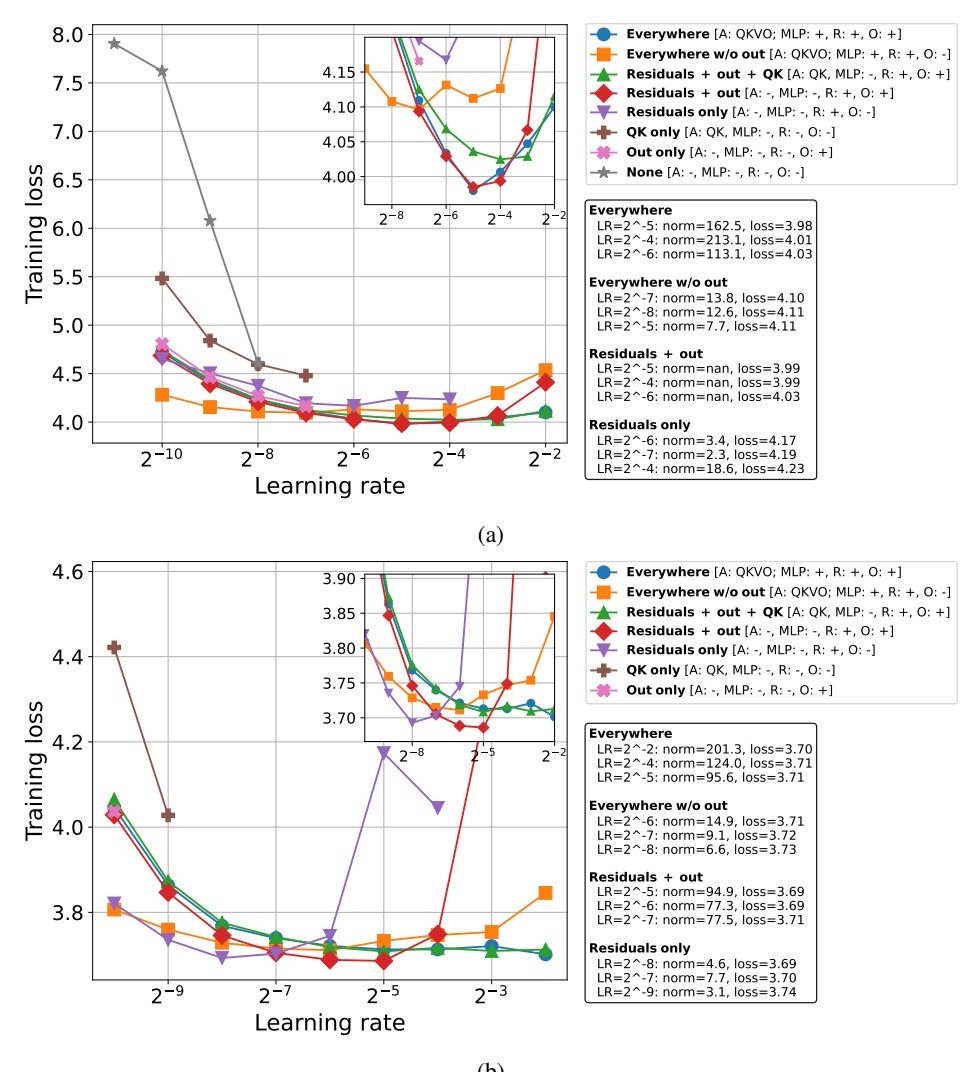

(a)

(b)

Figure 13: **Learning rate scan for various layer normalization strategies for the proxy model and the Scion optimizer: (a)** without momentum $\alpha = 1.0$, **(b)** with momentum $\alpha = 0.1$. All runs share the same batch size $B = 256$ [sequences] and the same learning rate schedule: no warmup, constant phase ($2^{35}$ tokens), linear decay to 0 for 20% of the total horizon ($0.25 \times 2^{35}$ tokens). Learning rate plotted on the X axis corresponds to the learning rate of the constant phase. Markers which are not present on the plot mean that the training diverged.

Notations in the legend corresponds to normalising 1) for the attention block (A) QKV: outputs of the QKV projection matrices, O: inputs to the output projection matrix; 2) MLP: inputs to the last linear layer; 3) residuals (R): inputs to both attention and MLP blocks; 4) Output layer (O): inputs to the final layer projecting the model dimension onto the vocabulary. Additional legend block shows the final norm ($\|W_{\mathrm{out}}\|_{\mathrm{RMS}\rightarrow\infty}$) value and training loss for the top-3 optimal learning rate runs within each normalization strategy.

### A.15 ABLATIONS ON FIG. 2A

#### A.15.1 MOMENTUM & LEARNING RATE DECAY

In this set of experiments we set momentum to 0.1 (which is by default disabled in the main text) and firstly run the same horizon scaling experiment for the proxy model (69M parameters) with the constant learning rate schedule and evaluate at the same horizons $D = \{2^{31}, 2^{33}, 2^{35}, 2^{37}\}$ as Fig. 2a. The results are presented in Fig. 14a. Here we perform loss smoothing in the same way as for the no-momentum scenario, but do not perform the fitting, i.e. for each batch size we take the optimal norm from the empirically best performing learning rate run. We find that the curves look more like "blobs", where multiple batch sizes give almost the same performance and are centered around the optimal norm (which also transfers across horizons). Also the loss difference between horizons is not well-pronounced as in the no-momentum scenario.

Then, we add learning rate decay, where we start from checkpoints of the horizons specified above, assume that that constitutes 75% of the total horizon, and linearly decay learning rate to 0 for the rest 25%. Likewise, we smooth loss values and take optimum value per batch size across empirical ones on the learning rate grid. In Fig. 14b we see that there is potentially a slight drift of the optimal norm with horizon scaling. However, after examining individual scans (Fig. 18) we surprisingly found that for long horizons the learning rate decay smooths out the norm optimum: a broad range (factor $\times 4 - 8$ in norm) of learning rates results in the same loss. Hence, there is no longer a single optimal norm, but rather a sizeable range, indicating that learning rate decay significantly reduces norm sensitivity. Therefore, we conclude that the seaming drift in Fig. 14b is not significant.

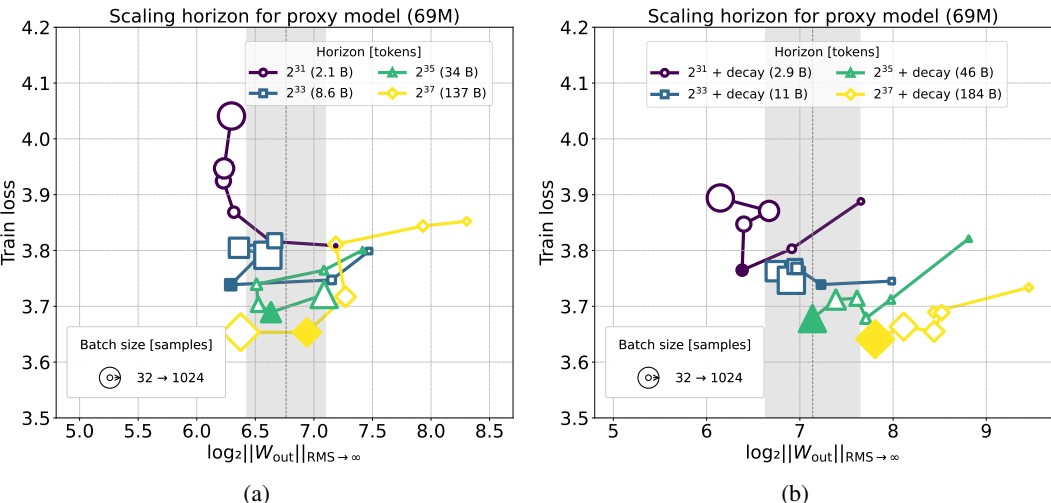

(a)                                                          (b)

Figure 14: **Same as Fig. 2a but with momentum = 0.1. (a)** Without learning rate decay. **(b)** With linear learning rate decay to 0 for extra 25% of the total horizon.

### A.15.2 NORM CHOICE

In this Section, we ablate if it is only the output layer norm that induces norm transfer. We replot Fig. 2a, with loss smoothing and without fitting, but now where we use $\|.\|_{\mathrm{RMS}\to\mathrm{RMS}}$ norm of the output or $\|.\|_{1\to\mathrm{RMS}}$ of the input layers instead default $\|.\|_{\mathrm{RMS}\to\infty}$ of the output layer. We observe in Fig. 15 (see also individual norm scans in Fig. 19) that interestingly both norms induce norm transfer.

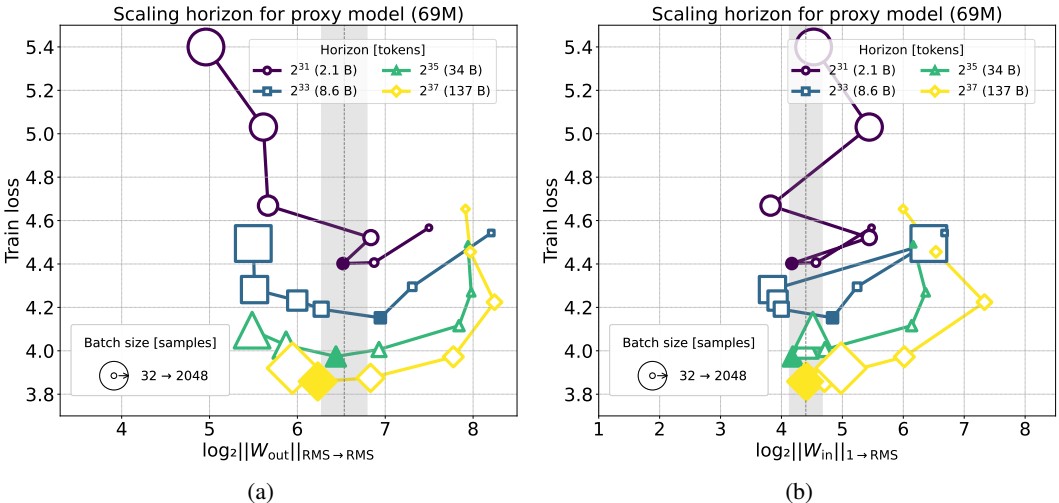

(a)                                                     (b)

Figure 15: **Same as Fig. 2a but with $\|W_{\mathrm{out}}\|_{\mathrm{RMS}\to\infty}$ norm for the X-axis changed to: (a)** $\|W_{\mathrm{out}}\|_{\mathrm{RMS}\to\mathrm{RMS}}$ (output layer). **(b)** $\|W_{\mathrm{in}}\|_{1\to\mathrm{RMS}}$ (input layer).

### A.16 LEARNING RATE LAYOUT FOR ADDITIONAL BATCH SIZES AND HORIZONS

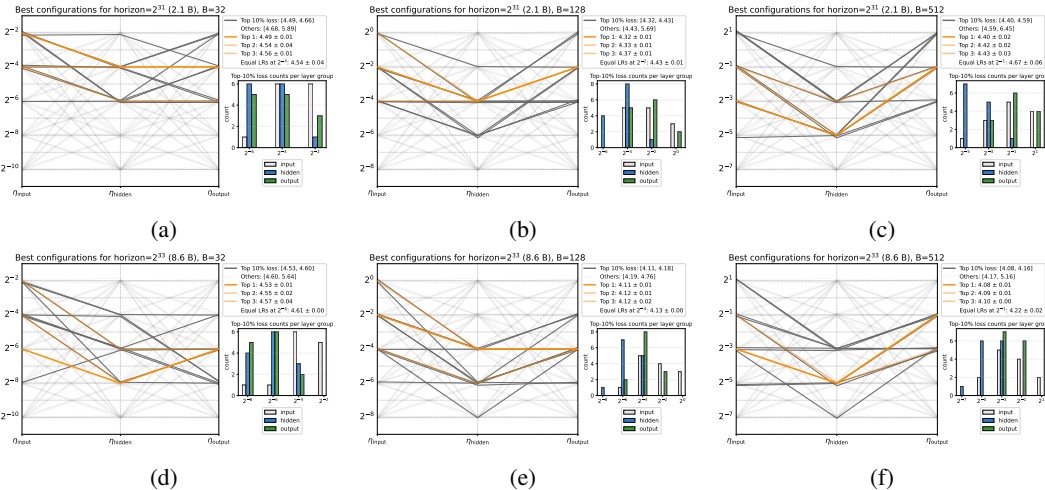

(a)                                      (b)                                      (c)

(d)                                      (e)                                      (f)

Figure 16: **Extended version of Fig. 4a with additional batch sizes and horizons.** Top (bottom) row: $D = 2^{31}(2^{33})$ token horizons. Batch sizes, in samples: $B = 32$ (left), $B = 128$ (middle), $B = 512$ (right). Performance is averaged across random seeds as described in Appendix A.2. Note that the optimal $B^*$ is 128 for both $D = 2^{31}$ and $D = 2^{33}$ according to Fig. 3b.

## A.17 INDIVIDUAL NORM SCANS AND FITS

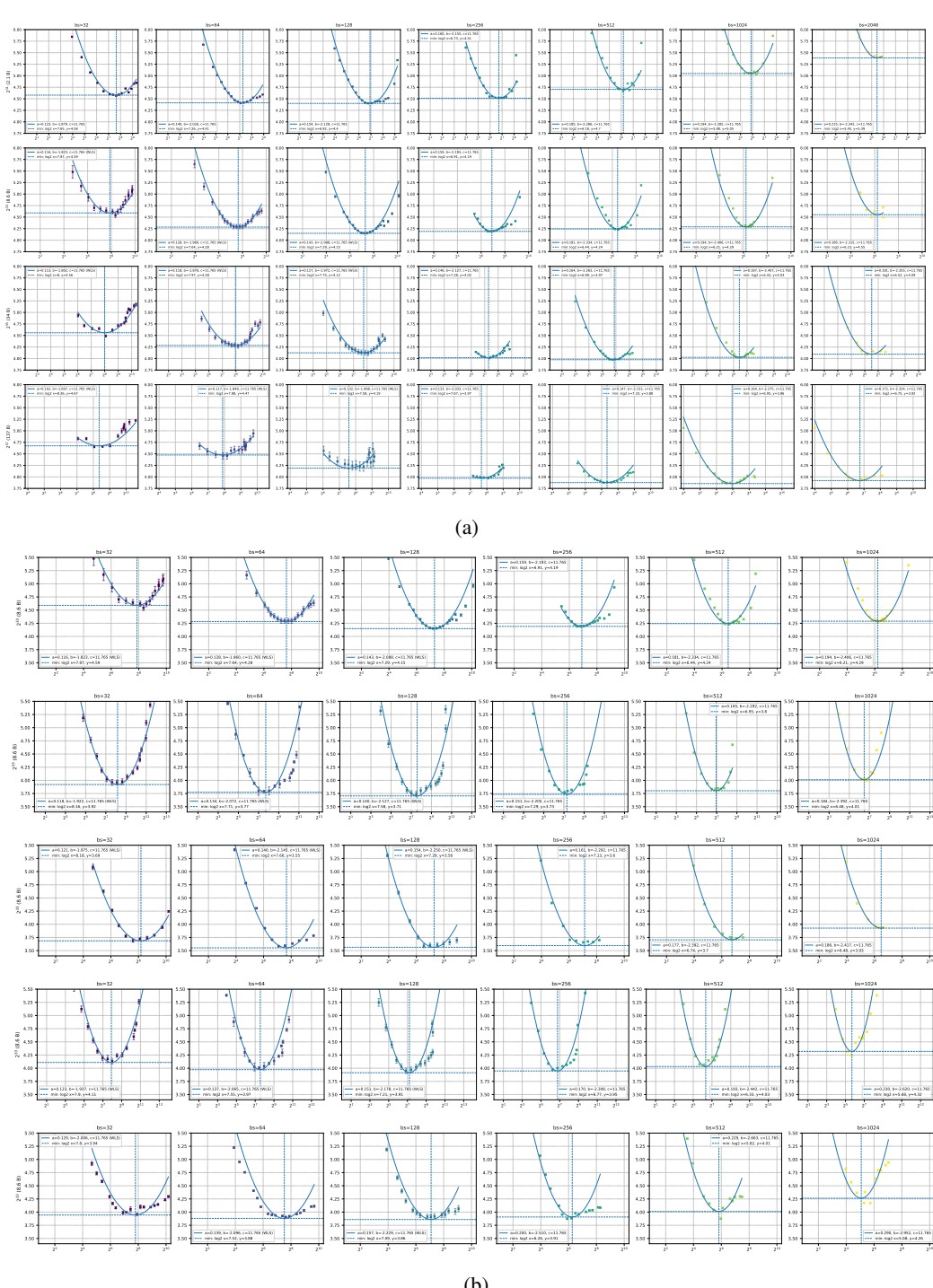

Figure 17: **Individual norm scans for various batch sizes** $B$ **(columns), across various horizons** $D$ **in (a), across various models in (b) (rows).** We plot train loss (Y-axis) against the output layer operator norm $\|\boldsymbol{W}_{\text{out}}\|_{\text{RMS}\to\infty}$, where each point corresponds to a different learning rate run and error bars correspond to loss smoothing variance (see Appendix A.4). The best-loss point for each $(B, D)$ is pinpointed with the blue dashed line, fitted curves are shown with blue solid lines. These fit results are used for: **(a)** Fig. 2a and Fig. 3b, **(b)** Fig. 2b, from top to bottom rows: proxy, $\times 4$-width, $\times 12$-width, $\times 8$-depth, $\times 32$-depth.

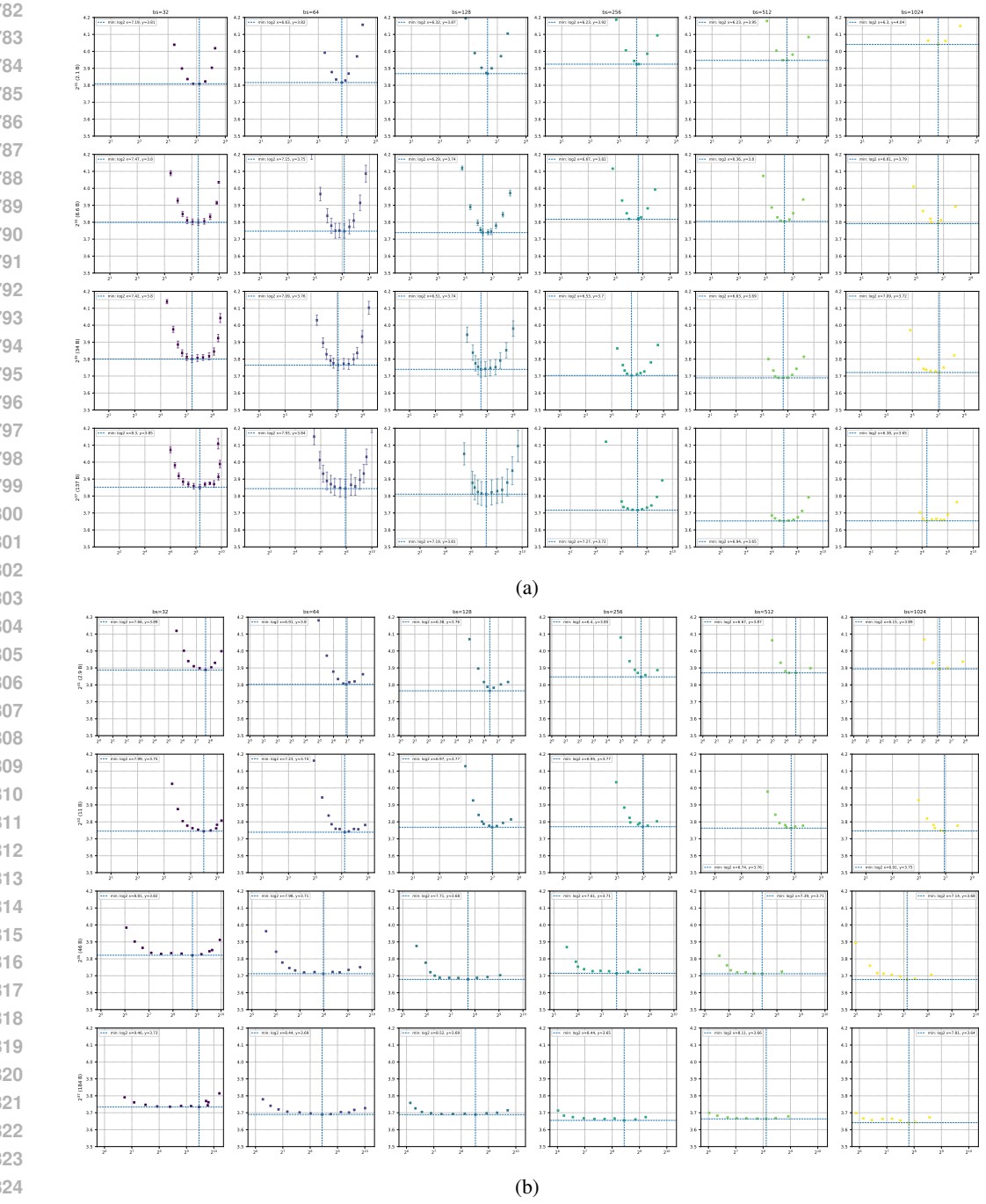

Figure 18: **Individual output norm** $\|W_{\text{out}}\|_{\text{RMS}\to\infty}$ **scans for various batch sizes** $B$ **(columns) across various horizons** $D$ **(rows).** (a) with momentum = 0.1, no learning rate decay. (b) with momentum = 0.1, with learning rate decay linearly to 0 for 25% of total horizon.

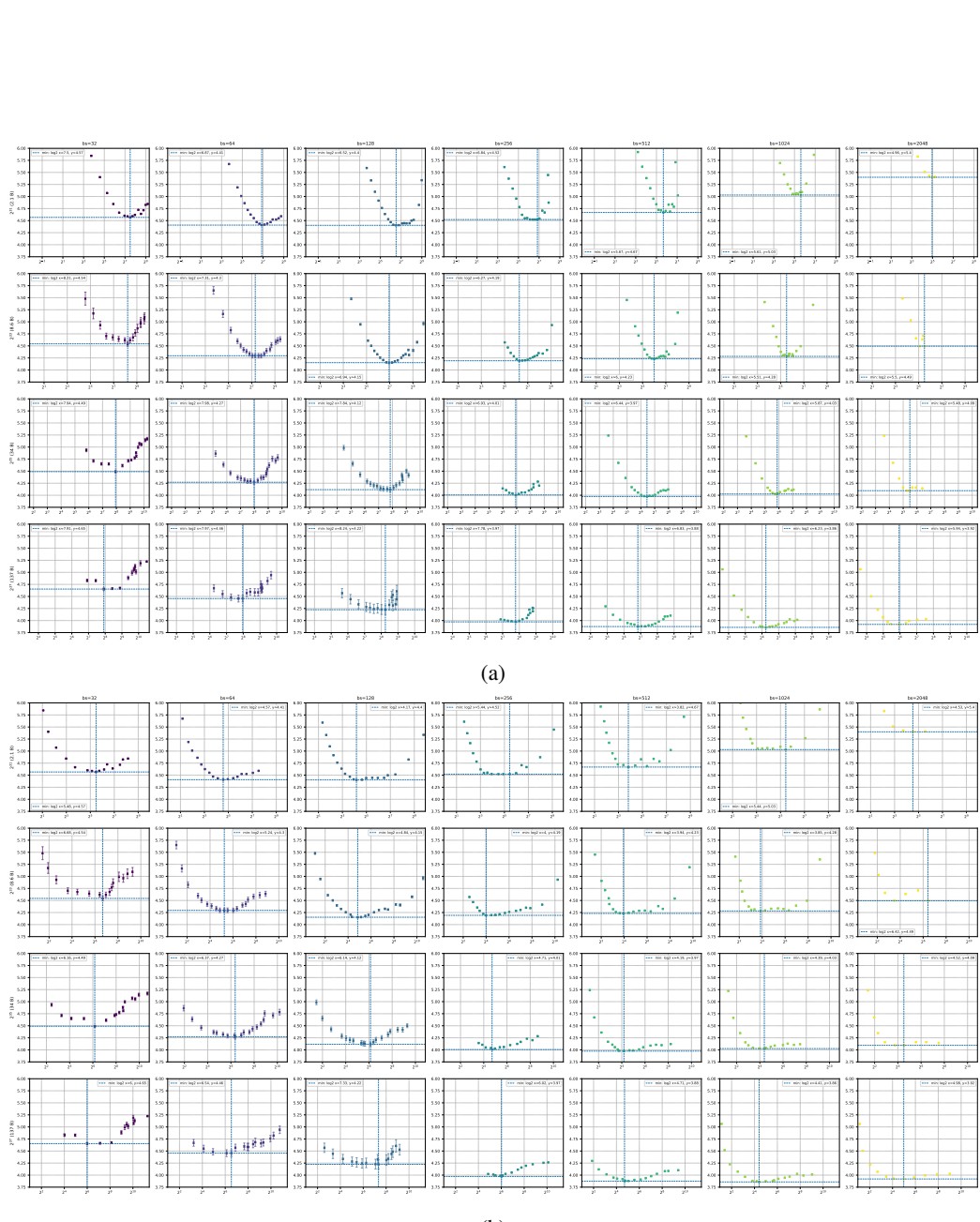

Figure 19: **Individual norm scans for various batch sizes** $B$ **(columns) across various horizons** $D$ **(rows). (a)** For $\|\boldsymbol{W}_{\mathrm{out}}\|_{\mathrm{RMS}\to\mathrm{RMS}}$ (output layer). **(b)** For $\|\boldsymbol{W}_{\mathrm{in}}\|_{1\to\mathrm{RMS}}$ (input layer).

