# OpenReview forum: "Optimal Scaling Needs Optimal Norm"
_ICLR.cc/2026/Conference — Submitted to ICLR 2026_

### Official Review · Reviewer_NKpP · 2025-10-26

**Soundness:** 2
**Presentation:** 2
**Contribution:** 1
**Rating:** 4
**Confidence:** 5

**Summary:**

This paper studies optimal tuning of LLMs as LR, model size, dataset size, and batch size vary.  Specifically, they consider tuning of norm-based optimizers such as Scion, where the operator norm of different layers is controlled as part of the optimization process.  Over a number of training runs, they identify that, at optimal settings of LR and B, the operator norm of the output layer is fairly consistent as both model and dataset size scale.  Further, they derive empirical power laws for optimal LR and B.  They also show that per-layer-group tuning of LRs can also help with Scion.

**Strengths:**

It's interesting to consider optimal hyperparameter scaling rules for norm-based optimizers, since this has previously mostly been done in AdamW, with and without maximal update parameterization.

Since norm-based approaches give new metrics that you can monitor, it is well-motivated to monitor them, and to derive insights from them.

I also don't mind the idea of stress-testing at a constant learning rate (even though this is definitely not state-of-the-art), as doing so can make things obvious that might not be when we use LR decay.  Similarly, it's perhaps okay to not use weight decay (but again, this deviates from the SOTA).

Some of the observations are definitely intriguing and thought-provoking.  For example, recent work is showing collapse/consistency in training loss curves (after normalization) across model sizes (https://arxiv.org/abs/2507.02119, https://arxiv.org/abs/2509.25087).  Here the operator norm of the output layer --- the first layer to get gradient from the loss --- is also consistent in a kind of normalized sense across scales as well.  I wonder if there’s likewise a connection between the output norm and scaling laws.  Especially since you use a constant LR, power laws for loss should hold at every step.

**Weaknesses:**

My overall view is that the way things are presented here is a bit of an overclaim, or a bit misleading for practitioners.  I have concerns about the real cause-and-effect, and whether the insights gained here are actionable.
- For example, we say that scaling is “governed” by the output layer operator norm, and we plot it as the independent variable, but it’s just something we observe, right?  We don’t scale or control it directly.  So I’m not sure how to use this, and secondly, correlation is not causation, right?
- In terms of actionable insights: the output norm seems like something you measure after a large-scale training run.  If you're training various runs, you wouldn't choose the one with lowest output norm, you'd choose the one with lowest loss, right?
- Moreover, even if your other hypotheses prove valid, how can you actually operationalize this?  How do you know the region of low norm sensitivity a priori, where you can exchange η for B?  Does this low-sensitivity region also transfer across scales?  “we cannot fully rely on the output norm as a guide to selecting optimal hyperparameters” – as a practitioner, how can you rely on it at all?

At least in the main paper, I would have liked more discussion of the case where you actually do use LR decay.  Do you still see output norm transfer?

Looking at the results as a whole, I'm not sure I *do* actually see norm transfer.  For example, Figure 1(a) and Figure 1(b) have different optimal output norms.  Perhaps they get even more different in other situations.  So it seems to be that output norm alignment is neither necessary nor sufficient.  So what *can* we really say about it???

More nitpicky:
- Methods section: How can we say that spectral norms (or MUP) “guarantee” hyperparameter transfer?  We already know that the amount of data affects the optimal HP settings, right?  So under what conditions do the HPs transfer?  With the same amount of data?  With a certain tokens-per-parameter ratio?  So in what sense is it a guarantee?
- “We briefly explain the idea behind each of them below.” – each of what?
- Tuning on a proxy model citing “(OpenAI et al., 2024; Gunter et al., 2024; Dey et al., 2024; Meta AI, 2025; Zuo et al., 2025)” – can you make the semantics of these citations more clear?  Like, does GPT-4 use MUP, and if not, what does it tune on a proxy model specifically?
- Why do we italicize “a single optimal batch size” in 3.2?  Is that surprising or notable somehow?
- Typo: “Later, a deeper understand* has been built”

**Questions:**

- Why isn’t there the same optimal output norm in Figure 1(a) and 1(b)?  One curve (d=256, D=2^33) is even plotted on both plots, right?
- What is the significance of RMS-norming the inputs to all layers?  Do we still need the same control on the operator norm?  Does it affect, e.g., how LR correlates with output norm?  You mention this might be one reason we get depth transfer --- could it be a reason we get norm transfer???
- Can we explain via theory, or even via intuition, why different layers would have different optimal LRs, when using norm-based optimizers?

---

> ### Author Response · Authors · 2025-11-19
>
> Thank you for the very careful review! Please find first our answers to the questions below:
> - *Why isn’t there the same optimal output norm in Figure 1(a) and 1(b)? One curve (d=256, D=2^33) is even plotted on both plots, right?*
>
>     We would like to point that the notion of “same” is not absolute, since we are dealing with random quantities subject to uncertainties (due to random seed and precision, mostly affecting orthogonalisation in the NS5 method). We did our best to methodologically reduce them (with e.g. fitting and averaging, see Appendix 4), but remaining effects will always be there. In this particular example, 2^33 curve is the only one from Fig. 1a which fluctuated slightly upwards from the mean across horizons (1 sigma away, +0.2 in norm value). And it was this one-off horizon for which we ran model scaled up runs of Fig 1b. The same +-0.2 variance is also observed in Fig 1b, so within uncertainties, we find the mean results to be consistent between the two figures.
>
> - *What is the significance of RMS-norming the inputs to all layers? Do we still need the same control on the operator norm? Does it affect, e.g., how LR correlates with output norm? You mention this might be one reason we get depth transfer --- could it be a reason we get norm transfer???*
>
>     That is indeed a very good point. We will run this ablation and update you shortly on what are the results. For now, we can point to https://arxiv.org/abs/2405.14813 and the notion of well-normedness introduced there and showing that normalising inputs is required for Linear layers (Appendix B). We would guess that without it, the signal propagation within the network is not “uniform” and becomes very heterogeneous, and so the gradient updates and hence scaling are affected. On the contrary, with layer norms everywhere, each layer operates on the “same” distribution of data. Personally, that would be also our candidate to explain both depth and norm transfer.
>
> - *Can we explain via theory, or even via intuition, why different layers would have different optimal LRs, when using norm-based optimizers?*
>
>     Intuitively, we would say that this is due to the heterogeneous structure of hidden layers. While input and output are just single Linear(), hidden is MLP (3 x Linear(), with gating and also not square matrices) +  Attention (QKVO+Softmax). Ideally, we would want to assign and tune a unique learning rate to each of those, but unfortunately this isn’t computationally feasible. So a single learning rate for hidden layers is a very rough quantity to cover the underlying complex structure.
>
>     We also wouldn’t say that disbalance in learning rate allocation is something specific to norm-based optimisers, rather it could be something related to the structure of computations and how normalisation is applied within them. In fact, it has been already found and studied with Adam from the sharpness perspective (https://arxiv.org/abs/2510.04202).
>
>     Besides, factor of x8 difference which we observe between input-output and hidden isn’t really “large”, at least comparing to the original work of https://arxiv.org/abs/2502.07529 (going up to O(1000)). For us the constant LR baseline isn’t significantly worse than the tuned one, so it may be that what we see is not a marginal effect but rather some second order correction.

---

> ### Author Response · Authors · 2025-11-19
>
> And now to address the questions raised in the Weaknesses section:
>
> - *For example, we say that scaling is “governed” by the output layer operator norm, and we plot it as the independent variable, but it’s just something we observe, right? We don’t scale or control it directly. So I’m not sure how to use this, and secondly, correlation is not causation, right?*
>
>     That is true that layer norms are observables which in our setup are not controllable. In that sense the usage of “governed” is misleading, we will remove it in the updated version of the paper.
>
>     But in this work we intentionally wanted to take an observing perspective and report phenomena which occur naturally during the training, without introducing explicit inductive biases. For example, norms can be controlled as e.g. studied in https://arxiv.org/abs/2507.13338, and that would be the next direction of studies. In our work we started just from observing the (output) norm, as we found it sensitive to “optimality”. The only indirect handle of control we have is tuning learning rate, which we are doing anyhow when scanning it.
>
> - *Moreover, even if your other hypotheses prove valid, how can you actually operationalize this? How do you know the region of low norm sensitivity a priori, where you can exchange η for B? Does this low-sensitivity region also transfer across scales? “we cannot fully rely on the output norm as a guide to selecting optimal hyperparameters” – as a practitioner, how can you rely on it at all?*
>
>     This is unknown a priori, same as optimal hyperparameters. For that, one has to scan them on a small scale to get the feeling of optimality and sensitivity range — and we believe there is no way around this. In that sense, optimal norm is not an exception and so should be scanned as well (in fact, it happens automatically as one tunes learning rate or batch size: effectively they are already scanning the norm). That is why we refer to the phenomenon as norm transfer, similar to hyperparameter transfer in muP. One has to perform a scan at small scale to then be able to “transfer” the learned insights on optimality and sensitivity to large scale.
>
>     Our main point is that looking at layer norms is extremely helpful in figuring out optimal scaling strategy (both a priori and during training). It does not provide full answers and full reliance, but already excludes a priori suboptimal hyperparameters and furthermore can instruct the practitioner during the training of the expected model performance.
>
> - *I would have liked more discussion of the case where you actually do use LR decay. Do you still see output norm transfer?*
>
>     Yes, we have this ablation in Appendix 12 and observe norm transfer. Furthermore, LR decay largely reduces norm sensitivity (see Fig. 13).
>
> - *Methods section: How can we say that spectral norms (or MUP) “guarantee” hyperparameter transfer? We already know that the amount of data affects the optimal HP settings, right? So under what conditions do the HPs transfer? With the same amount of data? With a certain tokens-per-parameter ratio? So in what sense is it a guarantee?*
>
>     With muP (or spectral condition) hyperparameter transfer is guaranteed assuming both fixed batch size and fixed number of steps. Unfortunately, in the original paper they claim transfer when those two hyperparameters are also scaled, but as was shown in (https://arxiv.org/abs/2410.05838) this is not the case.

---

> > ### Comment · Reviewer_NKpP · 2025-11-27
> >
> > Thanks a lot for your response, authors of "Optimal Scaling Needs Optimal Norm" paper!
> >
> > I did a fairly careful initial review of the original submission, and I would say that my original review still holds up well after the rebuttal.  I do think the presentation will be improved by not implying that a "unifying explanatory principle" has been found, and by not saying that optimal scaling is "governed" by optimal norm (but rather optimal configs seem to have the same operator norm across scale).  But yeah... still dunno whether that's actually true given Figure 2.

---

### Official Review · Reviewer_f6ch · 2025-10-28

**Soundness:** 2
**Presentation:** 4
**Contribution:** 2
**Rating:** 4
**Confidence:** 3

**Summary:**

This paper looks at how the optimal learning rate and batch size scale across model dimensions (width, depth) and dataset size for the Scion optimizer. In particular, they investigate how the norm of the final layer seems to be preserved across the optimal hyperparameter configurations, which they call norm transfer. These experiments are based on grid searches for relatively small (69M) Llama 3 models with additional normalization layers and generally no momentum or weight decay.

**Strengths:**

* The paper is well written and clearly has a lot of effort put into it.
* The topic of the paper (HP scaling across multiple dimensions) is interesting and relevant to the community.
* Good breadth of experiments and ablations.

**Weaknesses:**

* Some aspects of the experimental configuration choices may limit the relevance of the finding to typical training setups (e.g. no weight decay, no momentum, no learning rate schedule).
* It seems very likely that the norm transfer is simply a correlation with some other measure, rather than directly causing any interesting behaviors.
* I feel the paper somewhat lacks a clear practical takeaway. The norm observations can not directly be used (and may not hold with weight decay which is standard practice) and the HP scaling rules may also not hold for more typical training settings.

**Questions:**

* Could you clarify exactly which scaling rules and results hold in a more standard training setting (LR warmup + decay to zero, momentum, weight decay)?
* I think the paper would be stronger if you made an attempt to explain why the norm should transfer and when it does not. With weight decay it seems less likely, especially if you consider different (LR, WD) pairs. For the PyTorch AdamW version where the total decay scaling is (1 - LR * WD), different configurations with a constant LR*WD value often give similar final performance while affecting the norm differently.
* Do you believe your findings hold for other optimizers, e.g. AdamW?

---

> ### Author Response · Authors · 2025-11-19
>
> Thank you for your critical review! Please find our answers below:
> - *Could you clarify exactly which scaling rules and results hold in a more standard training setting (LR warmup + decay to zero, momentum, weight decay)?*
>
>     Actually, we started experiments from exactly this setup and quickly found that:
>
>     Firstly, it is not sensitive enough to make statistically significant claims. We report results with momentum and LR decay in Appendix A12: one can observe that there is norm transfer, but momentum and weight decay smooth a lot loss sensitivity to norm and learning rate variation. That makes scaling rules derivation meaningless, since we will be overwhelmed by experimental statistical noise. As for the warmup, the Scion optimizer doesn’t need, as discussed in the original paper (we also ablated this and saw that performance gets slightly worse with warmup).
>
>     Secondly, with two more hyperparameters we hit a curse of dimensionality, and additionally tuning them (momentum, weight decay) would explode our compute budget. So we had to compromise and stick to a minimalistic setup which would prove our research claims.
>
>     Lastly, a comment on weight decay (wd). It is known (https://arxiv.org/abs/2502.07529) that it constrains weight norms to 1/wd value (as we also show in Appendix A10). So tuning it would effectively result in clipping on the right in our main Figure 1a, without changing general trends and loss value (and actually compromising them due to over-regularisation at some point).
>
> - *I think the paper would be stronger if you made an attempt to explain why the norm should transfer and when it does not. With weight decay it seems less likely, especially if you consider different (LR, WD) pairs. For the PyTorch AdamW version where the total decay scaling is (1 - LR * WD), different configurations with a constant LR*WD value often give similar final performance while affecting the norm differently.*
>
>     The WD case you outlined is a so-called “coupled”, however there are known benefits of using “decoupled” version (https://docs.mosaicml.com/projects/composer/en/stable/api_reference/generated/composer.optim.DecoupledAdamW.html). And indeed in the coupled case there is degeneracy and it’s not obvious how to handle from the tuning perspective. Decoupled one is straight-forward to tune and in fact we used it in our setup for a dedicated WD ablation. From our ablations, we didn’t see any loss improvement from the weight decay, with its sole contribution being norm clipping (as also supported by theory). However, we would be very curious to explore it deeper in future work, as the current research frontier has also arrived at the point of studying such constrained optimisation.
>
> - *Do you believe your findings hold for other optimizers, e.g. AdamW?*
>
>     Not for SGD, but for Adam(W) yes, since it is also a norm-based optimizer — in the sense that raw gradients are normalised under a specific norm assumption. We are running Adam experiments now and will update you about the outcomes soon.

---

> > ### Comment · Reviewer_f6ch · 2025-11-24
> >
> > Thank you for the response. My main concerns unfortunately mostly remain. Overall the work feels like a very detailed and in some ways well executed study, but in the end it might lack concrete takeaways for a broader audience.
> >
> > On the practical side the norm observations can not be applied directly (as also mentioned by NKpP whose review I largely agree with) and do not seem to hold as well in a more standard experimental setups (at least the norm varies less if I understand correctly).
> >
> > On the theoretical or understanding side, I feel this lacks a mechanistic explanation for why the norm should / does transfer. With this we could perhaps get insights that we could confidently apply to other settings, but otherwise it is hard. For example, details like the choice of the norm-everywhere approach could matter significantly here, especially since it removes the learnable gains if I understand correctly. Without gains, the norm of the matrices strongly affect the size of the representations. This may matter significantly for the final loss for example and link the norm of the output matrix to well performing losses. Maybe this is not the case here, maybe there are other ways to control the final prediction norm (I am not sure from the manuscript which if any gains remain). But generally, without a good hypothesis that explains the phenomenon there could be various trivial causes like these, so I do not really know what to make of the information presented in the work.

---

### Official Review · Reviewer_HkV3 · 2025-10-28

**Soundness:** 3
**Presentation:** 4
**Contribution:** 3
**Rating:** 6
**Confidence:** 2

**Summary:**

This paper studies the optimal learning rate and batch size scaling in LLM training. Using the scion optimizer, they find that the optimal learning rate and batch size share the same operator norm of the output layer. This condition, however, is necessary and not sufficient, as different learning rates and batch sizes share the same operator norm. Through large-scale experiments, they provide empirical scaling laws for learning rate and batch size as functions of the dataset. They recover the known result that the optimal learning rate is proportional to the square root of batch size.

**Strengths:**

* Large-scale experiments
* The work unifies learning rate transfer and learning rate-batch size scaling laws
* The norm transfer result is intriguing (but I am unsure about the implications)

**Weaknesses:**

* The work is done on Scion optimizer, which is not well adopted in the field yet.
* The results are empirical, and it's unclear why norm transfer phenomena occur.

**Questions:**

* I would like to request that the authors help me understand the implications of this work --- as an optimal norm may not imply optimal performance, what is the impact of norm transfer?

---

> ### Author Response · Authors · 2025-11-19
>
> Thank you for your valuable feedback! Some comments about the weaknesses first:
>
> - *The work is done on Scion optimizer, which is not well adopted in the field yet.*
>
>     We want to emphasise that Scion belongs to the same family as the Muon optimizer, which is largely adopted now in frontier labs (e.g. https://moonshotai.github.io/Kimi-K2/) and lots of research has been published recently studying it and proving its benefits w.r.t. conventional optimizers (see citations in our paper). The difference is only that embedding layers are optimized in a more principled norm-based way with Scion, while Muon is a hybrid of norm-based optimisation and Adam. Therefore, we believe our observations are also transferable to Muon as it is being adopted by the community now. However, we are running a dedicated ablation with the Adam optimizer and will update you shortly once the results are ready.
>
> - *The results are empirical, and it's unclear why norm transfer phenomena occur.*
>
>     From the beginning, we largely viewed our work as a pure experimental observation, aiming at discovering new phenomena and presenting them as is, thus bringing new angles on understanding of LLM training at scale. That makes formal theoretical interpretation beyond the scope of our work, but with this work we definitely prompt for building such understanding by the community and we will follow up on it in our future work. However, we do provide additional reasoning behind it, see below.
>
> And now the question:
> - *I would like to request that the authors help me understand the implications of this work --- as an optimal norm may not imply optimal performance, what is the impact of norm transfer?*
>
>     Broadly speaking, with our work we aim to deepen the fundamental understanding of scaling from a single unifying principle. The least understood part so far is data scaling, since model scaling is largely solved by muP/modular norm/norm-based optimizers. From that perspective, we view norm transfer as a first step to unite model scaling with data scaling from a norm-base angle (setting aside that it’s a surprising discovery on its own).
>
>     Practically, norm transfer means that one necessarily has to keep the layer norms around the same value by adjusting hyperparameters accordingly, as they scale up — no matter model or dataset size (assuming correct model scaling framework). If the model doesn’t have the optimal norm, it is already not going to have optimal performance. It is in that sense we refer to the norm as “optimal”.
>
>     Then, “transfer” means that one can find the optimal norm value at small scale (both data and model), and then already know that this norm value has to be achieved for the target scaled-up run. Here, we are largely influenced by the notion of “hyperparameter transfer” introduced with muP, where one can find e.g. optimal learning rate on a small proxy model and then “transfer” it to a larger one to scale optimally.
>
>     It is rightly said that norm transfer isn’t sufficient, being only a necessary condition. Ideally, we would like to have a single both necessary and sufficient criteria which would provide us with a “scaling recipe” from a single fundamental “law of nature”. This is a clear limitation of our work, which we also outline in Sec. 5.
>
>     However, we do provide scaling rules for hyperparameters to scale optimally (without any fundamental underpinning), so practitioners can just use them without thinking too much. In fact, these scaling rules are all you need to scale optimally from the practical side and not the norm transfer, which only plays a “guidance” role from the pragmatical side. Importantly, we also show that at each dataset scale there is only one single optimal (LR, B) pair achieving optimal loss, with scaling rules interpolating between optimal hyperparameter values. This knowledge is crucial for practitioners, as it is still common to select batch size arbitrarily or as to maximise GPU utilisation.
>
>     For us as researchers though, norm transfer is more interesting and important, while the scaling rules are secondary. But we believe both hint towards this fundamental law (e.g. the rules look too nice as square-root and close to the muP rules, why?). So with our work we have effectively made half a step and are super curious what the community thinks about it and what could be the next step to solve the scaling puzzle (sadly, we don’t have all the answers for now).

---

### Official Review · Reviewer_y4hf · 2025-11-01

**Soundness:** 3
**Presentation:** 2
**Contribution:** 3
**Rating:** 4
**Confidence:** 3

**Summary:**

The paper investigates optimal hyper-parameter scaling for LLM training. The authors claim that the joint optimal scaling over model and dataset size is governed by a single invariant, the operator norm of the output layer. They study this optimal norm, specifically for Scion/Muon optimizer. The authors also mention that having an optimal norm is a necessary but not sufficient condition.

**Strengths:**

Strengths:
- The paper has a clear experimental methodology, with strong theoretical explanations about the approach.
- The paper tests norm transfer across a range of width, depth and data scales.
- The paper poses some very interesting questions and theoretical gaps for the research community towards the end of the main text.
- The authors release their Distributed Scion implementation, which can be helpful for the broader research community.

**Weaknesses:**

Scope for improvement:
- Since the paper only looks at the invariant optimal norm for the Scion/Muon optimizer, the applicability is narrow and cannot be generalized to other widely used optimizers.
- Most of the experiments in the paper are performed on a 69M parameter model, which is really small. The authors should shed some light on why they didn’t use bigger models which are more representative of the current SOTA model size. I am also curious how the optimal norm changes across different data regimes for a fixed set of models.
- The paper evaluates optimality solely through training loss (cross-entropy), without downstream task benchmarks. While training loss is standard for scaling law research, validating that norm-optimized configurations also optimize downstream performance would strengthen the claims. Even for small models (69M-1.3B), evaluating trends on standard benchmarks (e.g., HellaSwag, LAMBADA) would confirm that norm transfer reflects meaningful capability improvements, not just training dynamics artifacts. This is particularly important given the paper's claim of discovering a 'unifying principle' for optimal scaling.
- The anonymous github repo (https:// anonymous.4open.science/r/disco_iclr2026-E11D) seems empty, which raises some reproducibility concerns.

**Questions:**

Points 1, 2, 3 from weaknesses section.

---

> ### Author Response · Authors · 2025-11-19
>
> Thank you for your thoughtful review! Please find our replies to the questions below:
>
> - *Since the paper only looks at the invariant optimal norm for the Scion/Muon optimizer, the applicability is narrow and cannot be generalized to other widely used optimizers.*
>
>     We want to emphasise that Scion belongs to the same family as the Muon optimizer, which is largely adopted now in frontier labs (e.g. https://moonshotai.github.io/Kimi-K2/) and lots of research has been published recently studying it and proving its benefits w.r.t. conventional optimizers (see citations in our paper). The difference is only that embedding layers are optimized in a more principled norm-based way with Scion, while Muon is a hybrid of norm-based optimisation and Adam. Therefore, we believe our observations are also transferable to Muon as it is being adopted by the community now. However, we are running a dedicated ablation with the Adam optimizer and will update you shortly once the results are ready.
>
> - *Most of the experiments in the paper are performed on a 69M parameter model, which is really small. The authors should shed some light on why they didn’t use bigger models which are more representative of the current SOTA model size. I am also curious how the optimal norm changes across different data regimes for a fixed set of models.*
>
>     We believe this model size is already more than sufficient to measure the properties we are interested in. Since we look at the layer norms and their evolution with the training data size, by optimizer design we have strict theoretical guarantees that scaling up the model will result in exactly same layer norm values and exactly same training dynamics (as we elaborate in text and also check empirically on bigger models, see Fig. 1b). Therefore, from the research point of view measuring norm scaling with the dataset size for 69M or 100B parameter model is equivalent — except that it would require not ~100k A100-hours we have already spent (which is arguably a lot) but even more, without any research gains.
>
>     As for the data, this is indeed a very interesting question. We were also wondering how data distribution may affect observed phenomena, so we ran additional ablations with FineWeb-2 dataset, separately on Thai and Russian language partitions (we will add it to the updated version of the paper after the Adam ablation is done). Curiously, we don’t see any significant difference in optimal norm trends, with the optimal value remaining the same. We don’t exclude though that it requires significantly better precision to resolve difference across datasets (less than x2 grid step in batch size/learning rate), but so far the optimal norm value seems to be invariant.
>
> - *The paper evaluates optimality solely through training loss (cross-entropy), without downstream task benchmarks. While training loss is standard for scaling law research, validating that norm-optimized configurations also optimize downstream performance would strengthen the claims. Even for small models (69M-1.3B), evaluating trends on standard benchmarks (e.g., HellaSwag, LAMBADA) would confirm that norm transfer reflects meaningful capability improvements, not just training dynamics artifacts. This is particularly important given the paper's claim of discovering a 'unifying principle' for optimal scaling.*
>
>     It would be certainly beneficial to have more informative insights on scaling from the additional evaluation benchmarks. Unfortunately, there’s a drawback that at scale and scope of our experiments such evals would be extremely noisy and at the level of the random guess baseline (especially on 69M model) due to their statistical nature and lack of emergence in the model. Moreover, as we probe very fine-grained differences in loss due to precise hyperparameter tuning, we don’t expect them to be in any way reflected by standard benchmarks. That is why we decided to stick to the methodology adopted in the community when it comes to evaluating optimizers (https://arxiv.org/abs/2509.02046) or studying hyperparameter scaling (https://arxiv.org/abs/2503.04715): i.e. use loss (cross-entropy) which is the most informative and robust metric, additionally having clear fundamental research interpretation and being not prone to statistical and subjective biases of evaluation benchmarks.
>
> - *The anonymous github repo (https:// anonymous.4open.science/r/disco_iclr2026-E11D) seems empty, which raises some reproducibility concerns.*
>
>     The link works for us after removing the whitespace “ “ after “https://”

---

### Comment · Area_Chair_MTxG · 2025-11-27

Dear reviewers,

The authors have provided detailed responses to your reviews. I would appreciate if you could let both me and the authors know how these responses impact your assessment of the paper.

Best,

AC

---

### Author Response · Authors · 2025-11-27

Dear AC, all,

As also promised, we have run additional experiments with the Adam optimizer and various layer normalisation strategies to address the comments. The results are quite intriguing and we would say further underscore the message of the paper and our replies we already provided here in the discussion. We are almost done updating the manuscript with the corresponding plots and we will update you within the next 24h.

Kind regards,
Authors

---

### Author Response · Authors · 2025-11-28

Dear all,

It is very frustrating for us to hear that due to the recent news we cannot progress as before and you won’t be able to leave any further feedback. We therefore would like to greatly thank you for all the constructive comments and discussion so far. With this comment we update you on the experiments we have been running: namely, ablations on the Adam optimizer, normalization layers and dataset choice. We hope you find the results interesting and also addressing the points raised.

- **Adam optimizer** (see new paragraph in Sec. 3.2): we do observe norm transfer for both with (default betas) and without (betas=0) momentum versions. Interestingly, the momentum version shows a higher value of optimal norm, while no-momentum one (equivalent to signSGD), has exactly the same optimal norm value as no-momentum Scion. We believe that the presence of norm transfer for Adam even further strengthens and generalises the message of our paper. In fact, it is of no surprise to us: as was already shown in https://arxiv.org/abs/2409.20325, Adam (and especially it’s no-momentum version) can be naturally formulated within the norm-based view on optimization, which Scion is based on. So the shared patterns of norm transfer just further underscore the norm-based perspective.
- **Normalization layers** (see new paragraph in Sec. 3.2): firstly, for a fixed batch and horizon sizes we studied how various normalization strategies impact the optimal performance and the corresponding norm. In short, normalising inputs to attention and MLP blocks (”residuals norm”) is crucial to have a stable training, while addition of QK-norm, as known, largely reduces learning rate sensitivity. Our approach of norm-everywhere is in fact excessive: the same performance can be achieved with just keeping QK-norm, and normalizing inputs to every block and the model output layer. Important to note is the impact of the output norm layer: it is this layer which largely changes the optimal norm value. If removed, the optimal norm decreases by about x10 factor — the largest change among all of the ablations we have run.

    Secondly, we took the “residuals norm” configuration and studied its data scaling, i.e. ran multiple (learning rate, batch size) experiments for the proxy model across horizons up to 43B tokens. As a result, we also observe norm transfer, but as noted above, due to removal of the output norm layer at a noticeably lower norm value ~2^3 (vs. 2^7 for the main experiments). That points to the conclusion that norm transfer persists regardless of a specific normalisation strategy (as long as the training does not diverge).

- **Data distribution**: instead of the Nemotron-CC data (which is English only), we experimented with switching the data distribution on the language family side. We run the same main norm transfer experiment but using the Thai and Russian language partitions of the FineWeb-2 dataset. Expectedly, the norm transfer is still present, but interestingly we don’t see any significant optimal norm value change.

Finally, we updated the manuscript to incorporate the results of these ablations including minor editorial corrections, as well as with a new now Figure 1, to further illustrate to the reader how hyperparameter choice affects norm dynamics and so allows for controlling it.

Kind regards,

Authors

---

### Author Response · Authors · 2025-12-01

Dear all,

For further clarification, we would like to also address the latest comments of Reviewers NKpP and f6ch.

Firstly, for Reviewer NKpP:

- *I do think the presentation will be improved by not implying that a "unifying explanatory principle" has been found, and by not saying that optimal scaling is "governed" by optimal norm*

    We agree that the previous phrasing sounded misleading while we haven’t intended it to be so in the first place. In the updated paper version we clarified the relevant parts of the text and now phrase it as “unifying invariant” and “optimal scaling is conditioned” (further matching our main claims that norm transfer is only a necessary condition).

---

### Author Response · Authors · 2025-12-01

And now for Reviewer f6ch:

- *On the practical side the norm observations can not be applied directly <…> and do not seem to hold as well in a more standard experimental setups*

    As for the first part, the main idea which we have always emphasised about norm transfer is that it is not a silver bullet to solve scaling, but rather a guidance for the practitioner. On the practical side, with this work we bring the norm perspective onto the table and advocate that norms should be tracked similarly as one logs gradient norm during training. What norm transfer says is that if the norm is not within the optimal norm range once the training is finished, then one should already expect suboptimal performance. Furthermore, since norms evolve predictably, one can already predict throughout the training or importantly control via hyperparameter tuning what to expect at a given point in future (as we also illustrate in the paper, Sec. 3.1). Lastly, we specifically pulled together all the practical takeaways into summary boxes after each section, which we use ourselves in our large scale runs and which proved to be extremely helpful at large scale.

    As for the second part, we already elaborated in the earlier reply why we believe the contrary is the case. We did our best to include ablations with all the standard practices (and included additional ones thanks to the discussion here) and the observed results further support norm transfer. It is true, as we also point out in the main text, that usage of momentum and learning rate decay largely reduces the sensitivity to hyperparameter tuning. Therefore, the data points in these configurations are dominated by statistical noise and plots don’t necessarily have a clear distinct optimum (and we can’t really do anything about it to reduce this effect). That is why for the presentation of main results we intentionally opted for the no-momentum version, as the plots look cleaner and convey the message directly. But we always had the practical perspective in mind and therefore find this reduced sensitivity is even better from the practitioner side: it reduces the penalty of hyperparameter mistuning, while our observations are still applicable.

- *For example, details like the choice of the norm-everywhere approach could matter significantly here, especially since it removes the learnable gains if I understand correctly. <…> Maybe this is not the case here, maybe there are other ways to control the final prediction norm.*

    As described in Sec. 2.2, we don’t use learnable parameters in RMSNorm so there’s no gains in our setup. Besides, we initialise weights in a way that the corresponding norm, which is used to optimize a given layer, equals to 1. That makes the learning “equal” across the layers. Nevertheless, we further ablated various normalisation strategies and it indeed brings new insights on the observed norm transfer, as detailed in a dedicated general comment.

    As for the control of the norm, indeed there are multiple ways to control it. In our work, we illustrate it in Sec. 3.1, basically showing how learning rate can be indirectly used for this. Among the direct ways to control norms, we refer to a series of posts (https://leloykun.github.io/ponder/), showcasing e.g. methods of constrained manifold optimisation and various spectral clipping strategies.

- *On the theoretical or understanding side, I feel this lacks a mechanistic explanation for why the norm should / does transfer. With this we could perhaps get insights that we could confidently apply to other settings, but otherwise it is hard. <…>*

    We do acknowledge as limitation that so far we don’t have a complete understanding of observed phenomena. And we believe we don’t have to, for now, as research is a never-ending process of building up such understanding occurring step by step. We would be extremely happy ourselves to have the most fundamental ground truth zero-shot, but it turns out to not be easy within the scope of time we have been investigating and running experiments for this work.

    But it also doesn’t mean that we don’t care and don’t aim to have this understanding. On the contrary, for the paper we put our best effort to study and cover as much commonly used settings as possible, to illustrate that the norm transfer is omnipresent: from ablating momentum, LR schedules, architecture configurations, optimizers, normalization configurations (thanks for pointing this out!), datasets, and so on. So it looks like the phenomenon generalises quite well.

    That is why we strongly believe that making our work reach broader community is an important step towards really figuring out what is the reason behind our observations.


Thanks again for all your careful reviews! It is extremely unfortunate that we won’t be able to hear any comments back.

Kind regards,

Authors

---

### Meta-Review · Area_Chair_4j3k · 2026-01-05

**Summary:**

This paper studies how to transfer optimal learning rate and batch size as model size and dataset size scale, focusing on Scion and also discussing Adam. The central empirical claim is that, across a wide range of token horizons and with respect to width and depth scaling, the optimal learning rate and batch size configuration tends to result in a similar operator norm of the output layer (termed "norm transfer"). The paper also reports fitted scaling rules for optimal learning rates and batch sizes under horizon scaling, proposes per-layer group learning rates (with the output layer described as particularly sensitive), and releases a distributed Scion implementation along with extensive logs. Reviewers generally found the observation intriguing and the experimental effort substantial.

The revision and discussion partially address key concerns. In particular, the authors tone down some overly broad positioning and provide or promise additional evidence beyond the original Scion focus, including Adam experiments and normalization ablations. This is a good direction: framing the contribution more narrowly around the studied optimizer family and model and being explicit that the result is an empirical regularity, rather than a general explanatory principle, makes the paper more accurate and easier to evaluate. Even with these improvements, the main question is whether the observation yields actionable guidance or broader insight beyond a diagnostic tied to the chosen training stack.

**Reviewer Concerns:**

The main strengths emphasized by reviewers were the breadth of experiments and ablations, clear presentation, and the release of tooling and logs. The main weaknesses were uncertainty about causality, limited practical usefulness given the necessary but not sufficient framing, and concerns about external validity in standard training setups that include momentum, learning rate schedules, and weight decay. Several reviewers also flagged that parts of the initial framing read as overly strong.

The rebuttal enhances the framing and introduces new evidence suggesting that similar behavior may occur for Adam under certain normalization and dataset changes, while also acknowledging that sensitivity can decrease in more standard configurations. What remains outstanding is a decisive step that turns the observation into a reliable tool: either an intervention that explicitly shows controlling the output norm predictably improves outcomes, or a clear procedure that reduces hyperparameter search costs in realistic settings. Clearer positioning relative to established lines of work on hyperparameter scaling and batch size effects would also help clarify what is fundamentally new in this context.

**Reviewer Scores:**

Reviewer y4hf raised several concerns. The additional experiments with Adam additions address the optimizer scope concern, but do not fully resolve the broader significance and external validity issues, so I would expect y4hf to remain at 4 (or increase to 6 at most). Reviewer HkV3 was overall more positive but explicitly unsure about implications and expressed low confidence. I would expect HkV3's score to be 4 or 6. Reviewer f6ch focused on realism of the experimental setup and correlation versus causation; the rebuttal acknowledges these points and adds context, but not a decisive causal or practical resolution, so I would expect f6ch to remain at 4. Reviewer NKpP commented that they read the rebuttal and decided to keep their score.

---

### Decision · Program_Chairs · 2026-01-26

Reject